EMBO
Molecular Medicine

# Mutant *CTNNB1* and histological heterogeneity define metabolic subtypes of hepatoblastoma

Stefania Crippa[1,†], Pierre-Benoit Ancey[1,†], Jessica Vazquez[1], Paolo Angelino[2], Anne-Laure Rougemont[3], Catherine Guettier[4], Vincent Zoete[5], Mauro Delorenzi[2,6], Olivier Michielin[5] & Etienne Meylan[1,*] 

## Abstract

Hepatoblastoma is the most common malignant pediatric liver cancer. Histological evaluation of tumor biopsies is used to distinguish among the different subtypes of hepatoblastoma, with fetal and embryonal representing the two main epithelial components. With frequent *CTNNB1* mutations, hepatoblastoma is a Wnt/β-catenin-driven malignancy. Considering that Wnt activation has been associated with tumor metabolic reprogramming, we characterized the metabolic profile of cells from hepatoblastoma and compared it to cells from hepatocellular carcinoma. First, we demonstrated that glucose transporter *GLUT3* is a direct TCF4/β-catenin target gene. RNA sequencing enabled to identify molecular and metabolic features specific to hepatoblastoma and revealed that several glycolytic enzymes are overexpressed in embryonal-like compared to fetal-like tumor cells. This led us to implement successfully three biomarkers to distinguish embryonal from fetal components by immunohistochemistry from a large panel of human hepatoblastoma samples. Functional analyses demonstrated that embryonal-like hepatoblastoma cells are highly glycolytic and sensitive to hexokinase-1 silencing. Altogether, our findings reveal a new, metabolic classification of human hepatoblastoma, with potential future implications for patients' diagnosis and treatment.

**Keywords** glucose transporter; glycolysis; mutant β-catenin; pediatric liver cancer

**Subject Categories** Cancer; Digestive System; Systems Medicine

## Introduction

Hepatoblastoma (HB) and hepatocellular carcinoma (HCC) are the first and the second most common pediatric malignant liver tumors representing about 1–2% of cancers in children. HB is detected in very young children between the ages of 2 months and 3 years, while HCC occurs most frequently in children between 10 and 16 years old and is the predominant type of adult liver cancer. Fetal and embryonal HBs represent the two main epithelial subtypes, together with small cell-undifferentiated, pleomorphic poorly differentiated, cholangioblastic and epithelial macrotrabecular patterns (Lopez-Terrada *et al*, 2014). Pure fetal with low mitotic activity has the most favorable outcome and may be treated by surgery alone (Czauderna *et al*, 2014). Components of the Wnt/β-catenin pathway are frequently mutated and overactive in solid malignancies promoting tumor development (Clevers & Nusse, 2012). In HB, a driving proto-oncogene, and the most recurrently mutated gene, with 50–90% frequency, is *CTNNB1* that encodes β-catenin (Koch *et al*, 1999; Cairo *et al*, 2008; Lopez-Terrada *et al*, 2009; Eichenmuller *et al*, 2014). *CTNNB1* mutations in HB are located at exon 3, in a region of β-catenin important for its degradation by the proteasome (Aberle *et al*, 1997). Hence, in-frame deletions or missense mutations within exon 3 are gain-of-function mutations, resulting in a degradation-resistant β-catenin protein that accumulates in the nucleus, binds the TCF/LEF transcription factor, and drives the activation of target genes. Interestingly, in HB, large deletions in *CTNNB1*, which encompass exon 3 and part of exon 4, were reported exclusively in pure fetal tumor histotypes, whereas *CTNNB1* mutations in embryonal HB are small mutations confined to exon 3 (Lopez-Terrada *et al*, 2009). Although it is unclear whether and how different deletions affect β-catenin activity, the differential gene expression profiles in HB subtypes raise the possibility of context and time-dependent activation of β-catenin, leading to enriched expression of Wnt and stem cell-related genes in embryonal tumors, and activation of hepatic differentiation program in fetal tumors (Cairo *et al*, 2008; Lopez-Terrada *et al*, 2009). Recently, Wnt signaling was shown to increase glycolysis through the upregulation of pyruvate dehydrogenase kinase 1 (*PDK1*) and other glycolytic genes (Pate *et al*, 2014). This highlights an

1 Swiss Institute for Experimental Cancer Research, School of Life Sciences, Ecole Polytechnique Fédérale de Lausanne, Lausanne, Switzerland
2 Bioinformatics Core Facility, SIB Swiss Institute of Bioinformatics, Lausanne, Switzerland
3 Division of Clinical Pathology, Geneva University Hospitals, Geneva, Switzerland
4 Department of Pathology, Hôpital Bicêtre, HUPS, Assistance Publique-Hôpitaux de Paris, INSERM U1193, Faculté de Médecine, Université Paris Sud, Paris, France
5 Swiss Institute of Bioinformatics, Lausanne, Switzerland
6 Ludwig Center for Cancer Research and Department of Oncology, University of Lausanne, Lausanne, Switzerland
*Corresponding author. Tel: +41 21 693 7247; Fax: +41 21 693 7210; E-mail: etienne.meylan@epfl.ch
†These authors contributed equally to this work

important contribution of the Wnt pathway for reprogramming the cellular energetics of tumor cells.

It is becoming increasingly appreciated that different tumor types exhibit different metabolic activities depending on the tissue of origin and oncogenic mutations. Indeed, differences in glucose and glutamine metabolism were recently uncovered between *MYC*- and *MET*-induced liver tumors and between *MYC*-induced liver and lung cancer (Yuneva *et al*, 2012). Hence, the identification of tumor subtypes that differ metabolically could help to guide new diagnosis and treatment approaches. In this study, we highlighted molecular and metabolic differences between HB and HCC and characterized the expression of metabolic genes in pediatric liver cancer. This led us to define two different metabolic subtypes of HB with particular diagnostic biomarkers.

## Results

### Molecular characterization of HB cell lines

In an effort to comprehend molecular differences between two cancers that originate from the same tissue, HCC and HB, we profiled four HCC (Huh-1, Hep3B, HLE, and HLF) and four HB (Hep-U2, Huh-6, HepG2, and Hep293TT) cell lines using RNA sequencing (RNAseq). Principal component and heatmap analysis showed that the cell lines clustered by pairs (Fig 1A and B). Specifically, the two most epithelial HCCs (Huh-1 and Hep3B), as defined by CDH1$^{high}$, VIM$^{low}$, and GLUT3$^{low}$ (Masin *et al*, 2014), clustered together, separately from the two most mesenchymal ones (HLE and HLF), as defined by CDH1$^{low}$, VIM$^{high}$, and GLUT3$^{high}$, which clustered together, too. Within the HB group, Huh-6 clustered with Hep-U2 and HepG2 with Hep293TT (Fig 1A). Gene expression analysis revealed signatures of cytokine signaling enriched in the HCC group. In contrast, the HB group was enriched for signatures of carbohydrate transport and metabolism (Fig 1C), suggesting HBs differ from HCCs in sugar uptake and usage.

For this reason, we determined whether the differences between the two groups of liver cancer cells (HCC and HB) could be associated with different utilization of genes involved in glucose metabolism. We found a strong enrichment for glucose transporter expression in HB cells (Fig 1D). Among them, *GLUT3* was robustly expressed in HBs compared to HCCs, with Huh-6 and Hep-U2 exhibiting the highest levels (Figs 1D and 2A, and Dataset EV1). We recently reported that *SLC2A3* encoding for GLUT3 is induced by ZEB1 during epithelial–mesenchymal transition (EMT) in tumor cells from non-small cell lung cancer (NSCLC) and HCC (Masin

*et al*, 2014). However, when we compared HLE, a mesenchymal HCC cell line (Masin *et al*, 2014), with our panel of four cell lines derived from human HB, *GLUT3* expression was stronger in each of the HB cells (Fig 2A). Unfortunately, we could not maintain Hep-U2 in culture so our data on this cell line is limited to gene expression analysis. Immunocytochemistry (ICC) confirmed the strongest GLUT3 protein expression and membrane localization in Huh-6, followed by HepG2, Hep293TT, and HLE (Fig 2B). Huh-1, an epithelial HCC cell line, was negative. We then decided to explore whether this difference in GLUT3 expression is linked to a differential glucose uptake and utilization in the seven tumor cell lines. First, we monitored glucose uptake and found Huh-6 displaying a very high glucose consumption compared to every other cells (Fig 2C). We further explored whether this difference in glucose uptake may have consequences on the metabolic program of the different cells. Glycolysis and oxidation were determined through measurements of the extracellular acidification rate (ECAR) and oxygen consumption rate (OCR), respectively. The HB cell lines displayed a high oxidative and a low glycolytic profile (Fig 2D). Importantly, Huh-6 cells had a particular profile within HB, with a strong glycolytic metabolism (Fig 2D). The Huh-6 glycolytic profile was further confirmed using an independent assay based on the oxidative activity of mitochondria, showing the high glycolytic rate of those cells compared to all other cell lines (Fig 2E). To better characterize the glycolytic potential of the different cell lines, we performed sequential addition of glucose and 2-deoxyglucose (DG) after 24-h glucose starvation. As a result, we found that Huh-6 exhibited the highest glucose response in comparison with all other tested cell lines (Fig 2F). Interestingly, we also observed a response to glucose in Hep293TT confirming the glycolytic potential of these cells that we previously found with MitoTracker (Fig 2E). In contrast, we did not observe a clear response to glucose in the other tested cell lines. To understand the different metabolic profiles within HBs, we tested whether the less glycolytic cells, HepG2 and to a lesser extent Hep293TT, may have a higher fatty acid (FA) oxidation capacity. We measured OCR after addition of palmitate–BSA or BSA alone (vehicle control) after 24 h in limited medium (no glucose, 1% FBS). Only HepG2 consumed palmitate, while there was no response for Huh-6 (Fig 2G). A trend for increased consumption in Hep293TT was also detected, although this did not reach significance. Hence, decreased glucose use indeed correlates with increased FA use in HB cells. To understand better the molecular basis for FA consumption in the different cell lines, we took advantage of our RNAseq data to monitor the expression of *CPT1A* (carnitine palmitoyltransferase 1A), responsible for exogenous fatty acid incorporation into mitochondria. This gene was expressed in fetal-like cells with HepG2

---

**Figure 1.  Gene profiling reveals enhanced carbohydrate uptake and metabolism in tumor cells from HB compared to HCC.**

A  Multidimensional scaling (MDS) plot of the expression levels derived from the RNAseq. Distance between sample labels indicates similarity. Samples form four distinct groups according to the four cell line types.

B  Heatmap of the enrichment scores from the single-sample GSEA. Each column of the heatmap shows a cell line, while the rows represent gene sets. The plotted gene sets are the top 50 differentially enriched sets in HCC versus HB cell lines ranked by FDR. Color scale in the heatmap represents scores standardized across rows.

C  Heatmap of the enrichment scores from the single-sample GSEA. Each column of the heatmap shows a cell line, while the rows represent gene sets. The plotted gene sets are those that have been found as significantly enriched or depleted in HCC versus HB cell lines at a false discovery ratio (FDR) < 0.05. Differential enrichment is assessed on the ssGSEA enrichment scores with the limma package. Color scale in the heatmap represents scores standardized across rows.

D  Expression levels from RNAseq of the different glucose transporters differentially expressed in all HB compared to all HCC cell lines. Data show means ± s.d. (*n* = 4). *P*-values were determined by Mann–Whitney test.

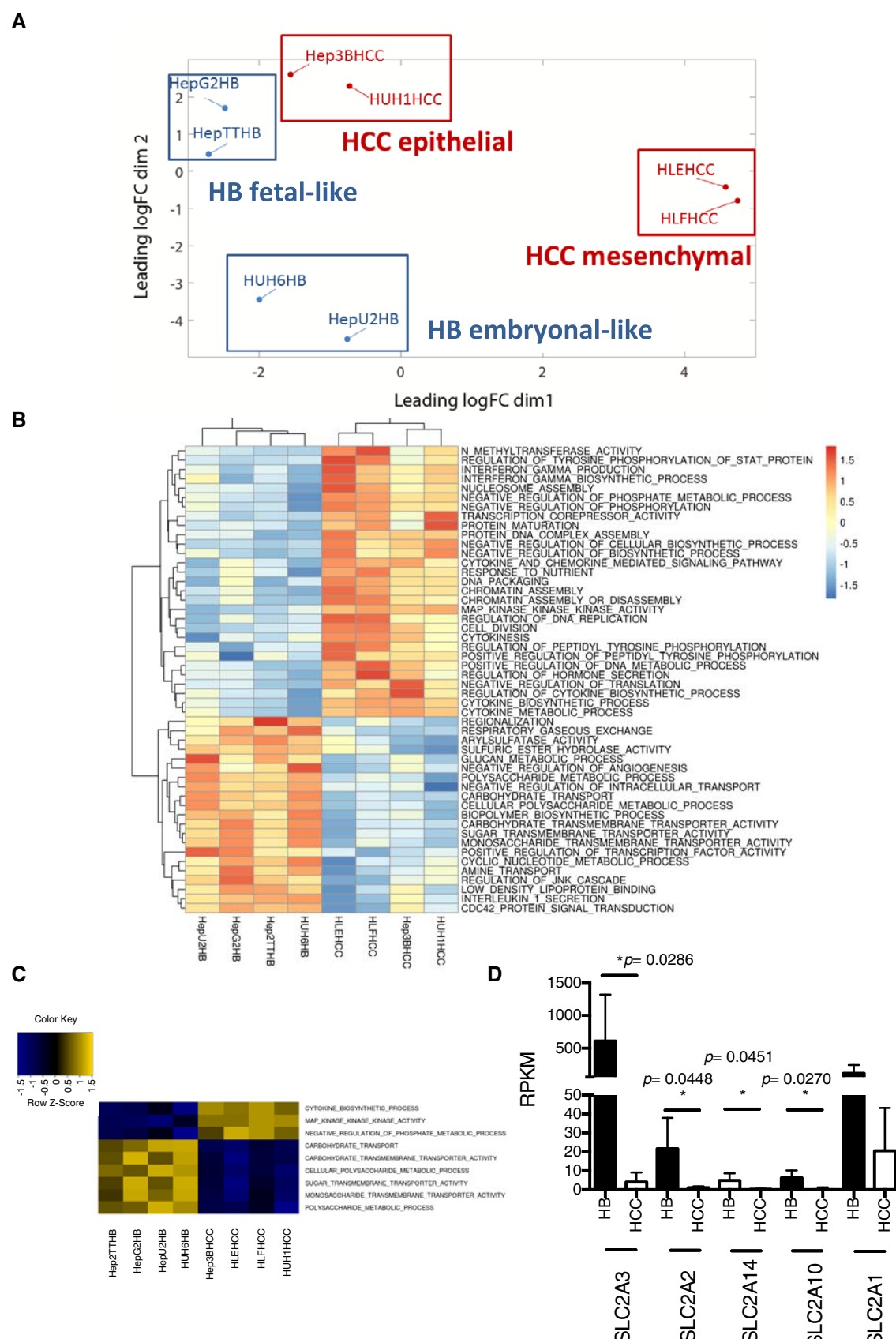

**Figure 1.**

Figure 2.   Identification of two HB metabolic subtypes.

A   Expression of *GLUT3* in the different cell lines. Data show means ± s.d. (n = 3). *P*-values were determined by Mann–Whitney test.
B   Immunocytochemistry of GLUT3 on the indicated cell lines. Dashed squares highlight a zoom on a cell. Scale bar: 20 µm.
C   2-Deoxy-d-[³H]glucose (DG) incorporation was measured in the indicated HB cell lines. Data show means ± s.d. (n = 5) of glucose uptake (nmol) normalized to protein concentration. Huh-6 values were significantly different (*P* < 0.05, Mann–Whitney test) to each other cell lines. The highest *P*-value is indicated.
D   Seahorse analysis of lactate production (ECAR) and oxygen consumption (OCR) in HB cell lines. Analysis was done on 10 measurements per sample and was performed 3 times. Data show means ± s.d. *P*-values were determined by Mann–Whitney test.
E   Flow cytometry analysis of the oxidative state (oxy) with MitoTracker Red (MTRed) and Green (MTGreen). Data are normalized to total mitochondrial mass (MTRed/MTGreen). Glycolytic state (gly) was calculated as 1 − (MTRed/MTGreen). Huh-6 values were significantly different (****$P$ < 0.0001) to each other cell lines. Data show means ± s.d. (n = 5). *P*-values were determined by Mann–Whitney test.
F   After 24 h of glucose starvation, 10 mM glucose was added to the wells followed by 2-DG to block glycolysis at a concentration of 50 mM. Sequential measurements of ECAR from 10 replicates after the injections were done. Data show means ± s.e.m.
G   After 24 h in limited medium, BSA-palmitate or BSA was added just before seahorse experiment; 5 sequential measurements of OCR from 10 replicates were done. Data show means ± s.d. *P*-values were determined by Mann–Whitney test.
H   After 24 h in limited medium, BSA-palmitate or BSA was added just before seahorse experiment in presence or in absence of 10 mM Etomoxir. Five sequential measurements of OCR from 10 replicates were done. Data show means ± s.d. *P*-values were determined by Mann–Whitney test.

displaying the strongest expression (Datasets EV1 and EV2), but was almost not expressed in embryonal-like Huh-6. In order to explore more precisely FA metabolism in the different HB cell subtypes, we used Etomoxir, a CPT1A inhibitor, which blocks exogenous fatty acid consumption. Only HepG2 oxygen consumption was affected by Etomoxir, revealing the highest exogenous fatty acid dependence in this cell type (Fig 2H).

To characterize the specific profile of Huh-6 within HBs, we examined broadly genes of glucose metabolism whose expression is, like that of *GLUT3*, significantly different between Huh-6 and the other HB cell lines. In total, we found nine differentially expressed genes implicated in glycolysis or the reverse biological process, gluconeogenesis. Specifically, in addition to *GLUT3*, three genes coding for enzymes of glycolysis, *HK1* (coding for hexokinase 1), *PFKP* (phosphofructokinase, platelet type), and *LDHB* (lactate dehydrogenase B), were significantly more elevated in Huh-6. In contrast, genes promoting gluconeogenesis, *PPARGC1A* (PGC-1α), an important liver gluconeogenesis transcriptional co-activator (Herzig *et al*, 2001; Yoon *et al*, 2001), *AQP9* (aquaporin-9), *GK* (glycerol kinase), and *G6PC* (glucose-6-phosphatase), were all less abundant in Huh-6 and Hep-U2 (Fig 3A, Appendix Fig S1A, Datasets EV1 and EV2), supporting the increased glycolysis of these cells compared to other HB cells. As notable exception, HepG2 and Hep293TT cells expressed *HK2* to a significantly higher level than Huh-6 and Hep-U2. This is interesting, as *HK2* is often overexpressed in adult solid tumors (Patra *et al*, 2013; Guo *et al*, 2015), which is compatible with a more differentiated state of those cells, while *HK1* expression is strong specifically in the developing mouse liver (Appendix Fig S1B).

Because of the singularity of Huh-6 cells within HBs for utilization of glucose and because HB is a β-catenin-driven cancer, we wanted to know whether the Huh-6 cell line could represent a different subtype of HB at the molecular level. Western blotting of β-catenin, frequently mutated in HBs, with an antibody recognizing the C-terminus, revealed two bands for HepG2 and Hep293TT, but only one for the other liver tumor cells including Huh-6 (Fig 3B). In contrast, a second antibody recognizing the N-terminus failed to detect the shortest form specific to HepG2 and Hep293TT, suggesting this variant lacks some amino acids located near the N-terminus (Fig 3B). Because HepG2 is known to carry a large *CTNNB1* deletion of exon 3 and part of exon 4, and Huh-6 a point mutation within

exon 3 (Koch *et al*, 1999), we hypothesized that Hep293TT carries a similar deletion than HepG2. To identify the exact *CTNNB1* mutations, we analyzed the exon coverage profile using the RNAseq reads. This revealed that (i) Huh-6 has a GGA to GTA point mutation at codon 34, resulting in a G to V amino acid substitution, (ii) Hep-U2 carries a small deletion confined to exon 3, and (iii) both HepG2 and Hep293TT have a large deletion, typical of fetal HB subtypes, encompassing exon 3 and part of exon 4 (Fig 3C). PCR-based genomic amplification confirmed the genomic deletions (Fig EV1A). Mutations in Hep-U2 and Huh-6 are reminiscent of embryonal HB. Additionally, real-time PCR and ICC showed a stronger expression of LIN28A, expressed in undifferentiated tissues and important for embryonic stem cells (Tan *et al*, 2014), in Huh-6 and Hep-U2 compared to HepG2 and Hep293TT. In contrast, *CLDN1*, which codes for claudin-1, a tight junction protein enriched in the fetal compared to the embryonal components of HB (Halasz *et al*, 2006), was more elevated in HepG2 and Hep293TT (Fig EV1B–D and Dataset EV1). Finally, we analyzed the top 200 differentially expressed genes between the two cell clusters using Gene Ontology (http://geneontology.org/). This revealed biological processes of development, morphogenesis, and embryogenesis from the genes up in Huh-6 and Hep-U2, compatible with a block in early development or undifferentiated stage. In contrast, the same analysis highlighted metabolic processes from the genes up in HepG2 and Hep293TT, reminiscent of the physiological function of differentiated hepatocytes (Datasets EV3 and EV4). Hence, Hep-U2 and Huh-6 classify as poorly differentiated embryonal-like HB, whereas HepG2 and Hep293TT represent more differentiated, fetal-like HB.

We then explored the impact of GLUT3 in the different cell lines with GLUT3 siRNA, which strongly decreased its expression (~80% decrease) (Fig 3D). Using this approach, we first found that GLUT3 was essential for glucose uptake (Fig 3E) in Huh-6 but not in HepG2. Second, we performed a glycolysis assay using seahorse. Huh-6 glycolytic capacity was altered by the reduction of GLUT3 while HepG2 was insensitive to GLUT3 decrease (Fig 3F). Finally, Hep293TT exhibited an intermediate response in absence of GLUT3 correlating well with their partial glycolytic phenotype observed with MitoTracker and seahorse experiments (Fig 2E and F).

In order to test the validity of our cell line-based classification in human samples, we used immunohistochemistry and stained 20

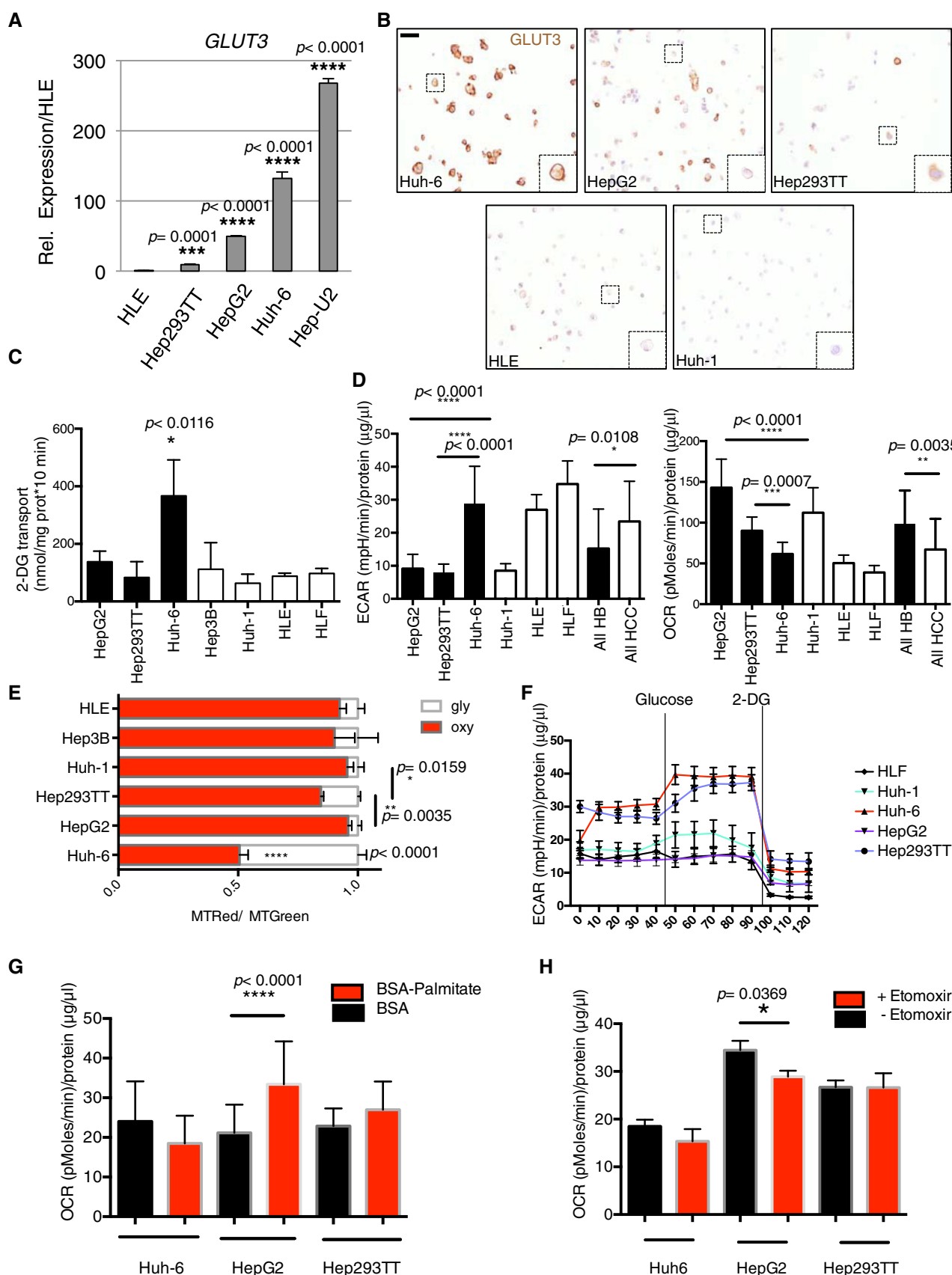

**Figure 2.**

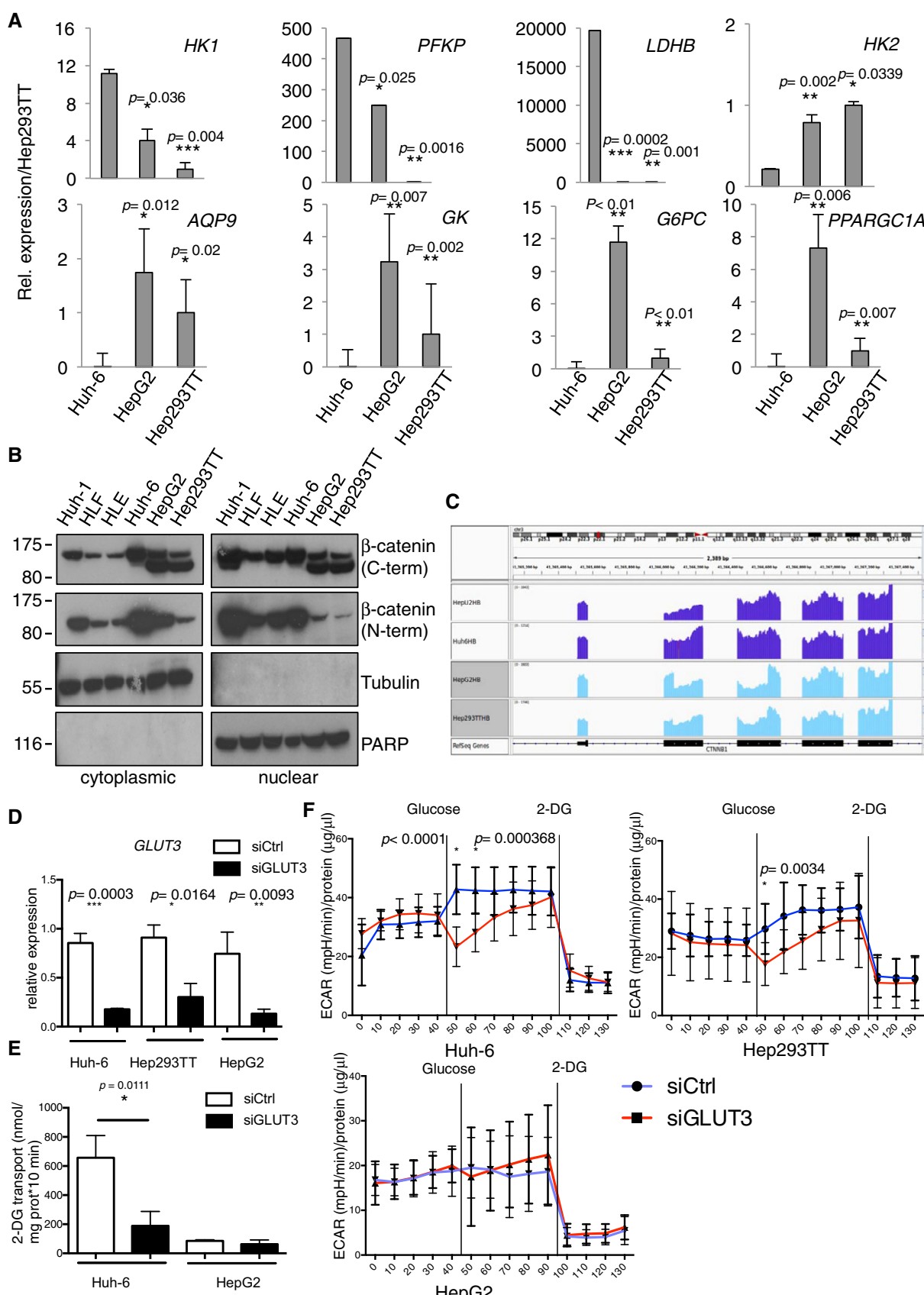

**Figure 3.**

**Figure 3.  The glycolytic profile of HB cell lines is correlated to β-catenin mutation.**

A   Real-time PCR using specific probes for glycolytic (*HK1*, *PFKP*, *LDHB*, *HK2*) and gluconeogenesis (*AQP9*, *GK*, *G6PC*, *PPARGC1A*) genes. Data show means ± s.d. (*n* = 3). *P*-values were determined by Mann–Whitney test.
B   Cytoplasmic and nuclear protein extracts isolated from the indicated cell lines were analyzed for β-catenin expression by Western blot.
C   Histograms of the RNAseq read counts for the *CTNNB1* region in the embryonal and fetal cell lines.
D   Real-time PCR on GLUT3 after siGLUT3 or siCtrl transfection. Data show means ± s.d. (*n* = 3). *P*-values were determined by Mann–Whitney test.
E   Glucose uptake assay in HepG2 and Huh-6 after transfection with siGLUT3 or siCtrl. Data show means ± s.d. (*n* = 5). *P*-values were determined by Mann–Whitney test.
F   Seahorse glycolytic assay on the three hepatoblastoma cell lines after transfection with siGLUT3 or siCtrl. Sequential measurements of ECAR from 10 replicates after the injections were done. Data show means ± s.d. Multiple *t*-test was used.

Source data are available online for this figure.

post-chemotherapy HB surgical specimens, which have been histologically classified and scored for their embryonal/fetal status by two pathologists (CG and ALR) using standard H&E or HPS staining (Dataset EV5). Because we identify GLUT3 as strongly expressed in embryonal-like cells, we assessed the ability of GLUT3 staining to distinguish embryonal and fetal tumors. While staining for GLUT3 was never observed in healthy liver, in tumor samples GLUT3 was exclusively expressed in a minority of the cells from embryonal foci (mild 1+ cytoplasmic reactivity, and mild 1+ or moderate 2+ membranous staining in 5/13 cases), or from the squamous component (mild 1+ or moderate 2+ staining in all three cases) (Figs 4A–C and EV2A–F). The latter finding prompted us to assess CK5/6 reactivity, a squamous epithelium and basal cell marker, in all HB cases displaying a histologically obvious squamous component, and/or reactivity to GLUT3. As expected, foci of squamous differentiation were reactive to CK5/6. Moreover, in one of four HBs with no obvious squamous differentiation, CK5/6 highlighted a basal-like cell component in areas reactive to GLUT3 (Fig EV2D–F). Importantly, the fetal cells did not stain for GLUT3 in any of the HBs with a fetal component (Fig 4A and C). Chemotherapy causes a decrease in GLUT3 expression (Watanabe *et al*, 2010), (Fig EV2G), suggesting GLUT3 expression is underestimated in post-chemotherapy HB samples. Thus, GLUT3 is a marker able to distinguish embryonal from fetal tumors.

### GLUT3 is a target gene of the Wnt/β-catenin pathway

Because HB is a β-catenin-driven malignancy, with c-Myc and YAP also contributing to disease development (Shachaf *et al*, 2004; Tao *et al*, 2014; Lehmann *et al*, 2016), we hypothesized one of them to be implicated in the sugar metabolic phenotype observed in HB cell lines. To test this possibility, we transfected Hep293TT, HepG2, and Huh-6 with a control siRNA, or with siRNAs targeting *CTNNB1* or *MYC*. *CTNNB1* but not *MYC* silencing resulted in a significant decrease in *GLUT3* mRNA (Fig 5A and Appendix Fig S1C). The expression of other glycolytic genes remained unchanged or increased upon transfection with siRNA targeting *CTNNB1* (Appendix Fig S2A). In a similar way, we assessed the potential role for YAP in GLUT3 regulation by both overexpression and silencing. We did not observe a strong impact of YAP alteration on GLUT3 expression reinforcing the specific role of β-catenin in GLUT3 regulation (Appendix Fig S2B and C). We decided to investigate further the link between β-catenin and *GLUT3*. First, we used two reporter constructs previously generated (Masin *et al*, 2014) of the promoter or the second intron of *GLUT3* placed before a minimal promoter driving luciferase expression. β-Catenin overexpression led to an

increase in the luciferase activity of the intron 2 but not the promoter construct (Fig 5B). To identify the region within this intron responsive to β-catenin, we generated four deletion constructs (F1-F4), and only F4 was induced (Fig 5C). We further divided F4 into F4a-c, which revealed that F4c was the most responsive to β-catenin (Fig 5D). Interestingly, a TCF4-binding site was predicted within this fragment (http://jaspar.genereg.net/) (Fig 5E), suggesting β-catenin acts as co-activator for the TCF/LEF transcription factor to stimulate *GLUT3* transcription; deletion of this sequence abolished the β-catenin-mediated increased luciferase activity (Fig 5E). To identify whether the Wnt/β-catenin signaling pathway directly promotes *GLUT3* gene induction, we performed chromatin immunoprecipitation (ChIP) on TCF4 and β-catenin followed by real-time PCR. As negative control, we used a region in Myc 5′UTR unresponsive to the Wnt/β-catenin (Bottomly *et al*, 2010). These results revealed a binding of TCF4, as well as β-catenin (most likely through TCF4), to the *GLUT3* intron 2, fragment F4, but not to F1, or to the promoter region (Fig 5F and Appendix Fig S3A and B). Interestingly, β-catenin binding was stronger in Huh-6. These findings suggest a link between the β-catenin different mutations and its ability to induce GLUT3, which may in turn be responsible for the strong glycolytic capability we observed in Huh-6.

### Long β-catenin deletion mutants do not interact with α-catenin

Because of the correlation between β-catenin mutation and GLUT3 binding, we investigated implications of the different *CTNNB1* mutations in fetal and embryonal HB cell lines. We found that genes associated with the canonical Wnt pathway are more expressed in the embryonal-like than the fetal-like HB cell lines (Dataset EV3). We reasoned that the extent of *CTNNB1* deletion could cause a loss of interaction with specific co-activators of Wnt-target genes. We first evaluated whether in fetal-like HB cell lines, the partial deletion of *CTNNB1* exon 4 could alter the capability of β-catenin to interact with BCL9 and BCL9L, which are established co-activators of the Wnt pathway important to specify a stem cell-like phenotype in cancer cells (Brembeck *et al*, 2004; Deka *et al*, 2010). We found that co-inhibition of *BCL9-BCL9L* only faintly reduced the expression of *GLUT3* in Huh-6 (Appendix Fig S4A) and we did not observe a stronger luciferase activity of *GLUT3* intron 2 upon transient overexpression of BLC9L together with β-catenin (Appendix Fig S4B). In agreement with this, the alpha-helix responsible for BCL9-BCL9L binding (Sampietro *et al*, 2006) is still present in the long deletion form of β-catenin. In marked contrast, we modeled that the residues encoded by exon 4 that are missing in fetal β-catenin mutants are

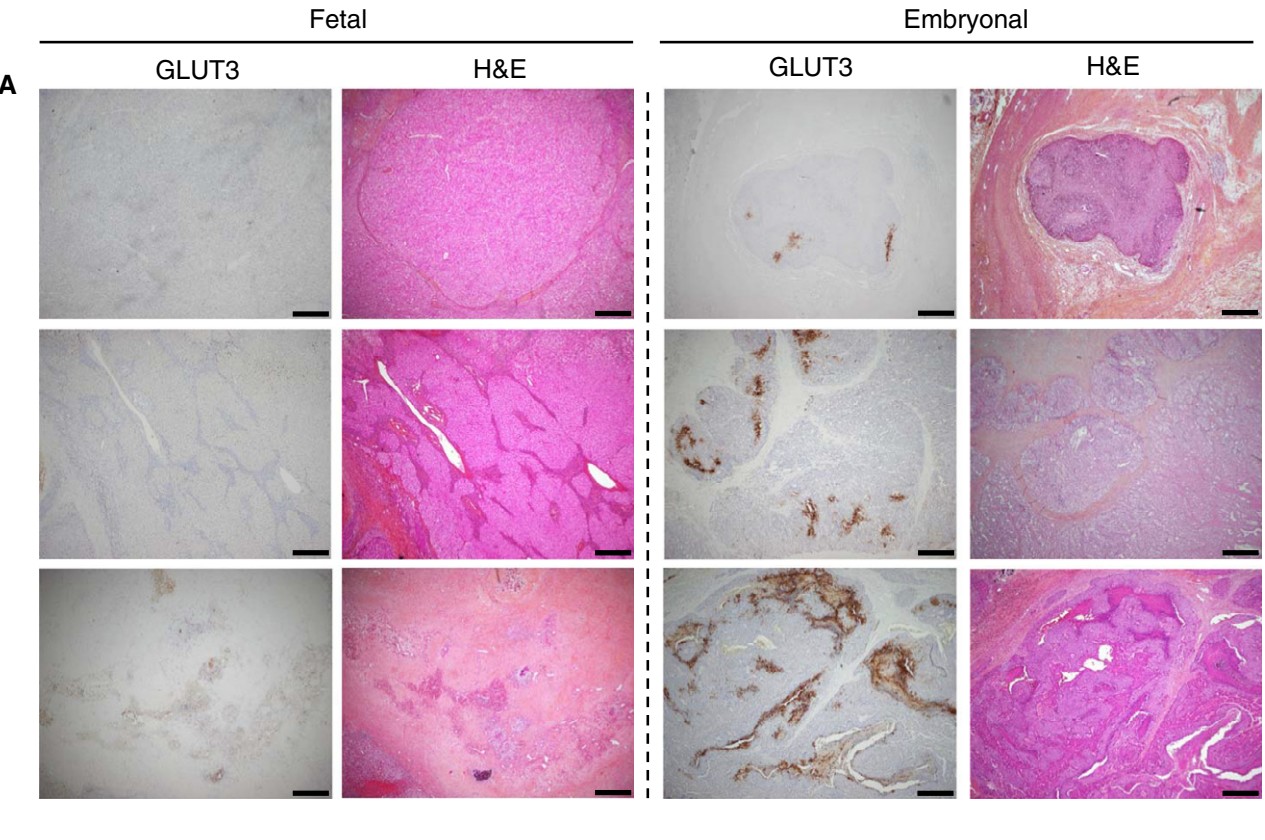

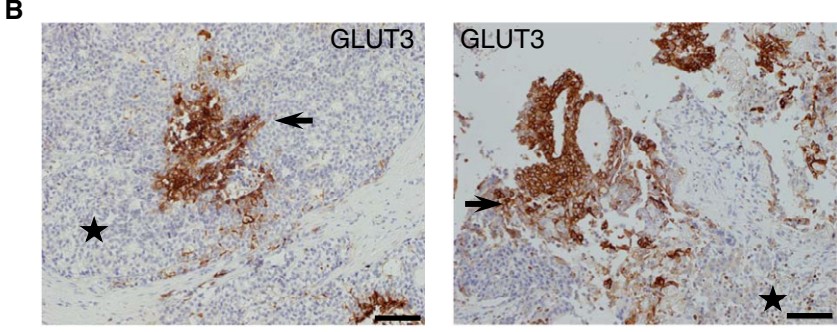

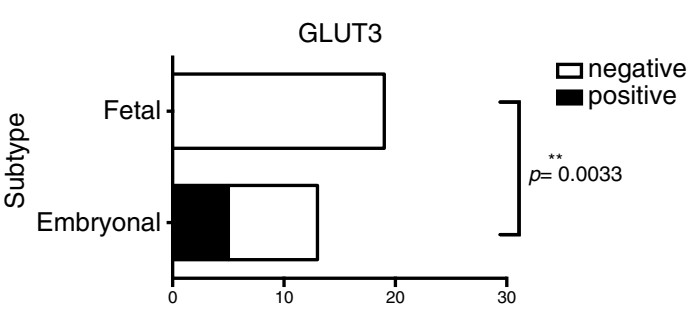

**Figure 4.  The embryonal component of human HB tumors stains positive for GLUT3.**

A   Representative H&E or GLUT3 staining of human HB tumor sections. Scale bars: 200 μm (original magnification ×40).

B   Representative GLUT3 IHC staining of two human HB tumor sections at high magnification showing the embryonal component positive for GLUT3 expression. Stars: fetal component; arrows: embryonal component. Scale bars: 100 μm (original magnification ×100).

C   Contingency table between the absence/presence of the staining and the embryonal/fetal status. Chi-square test was used.

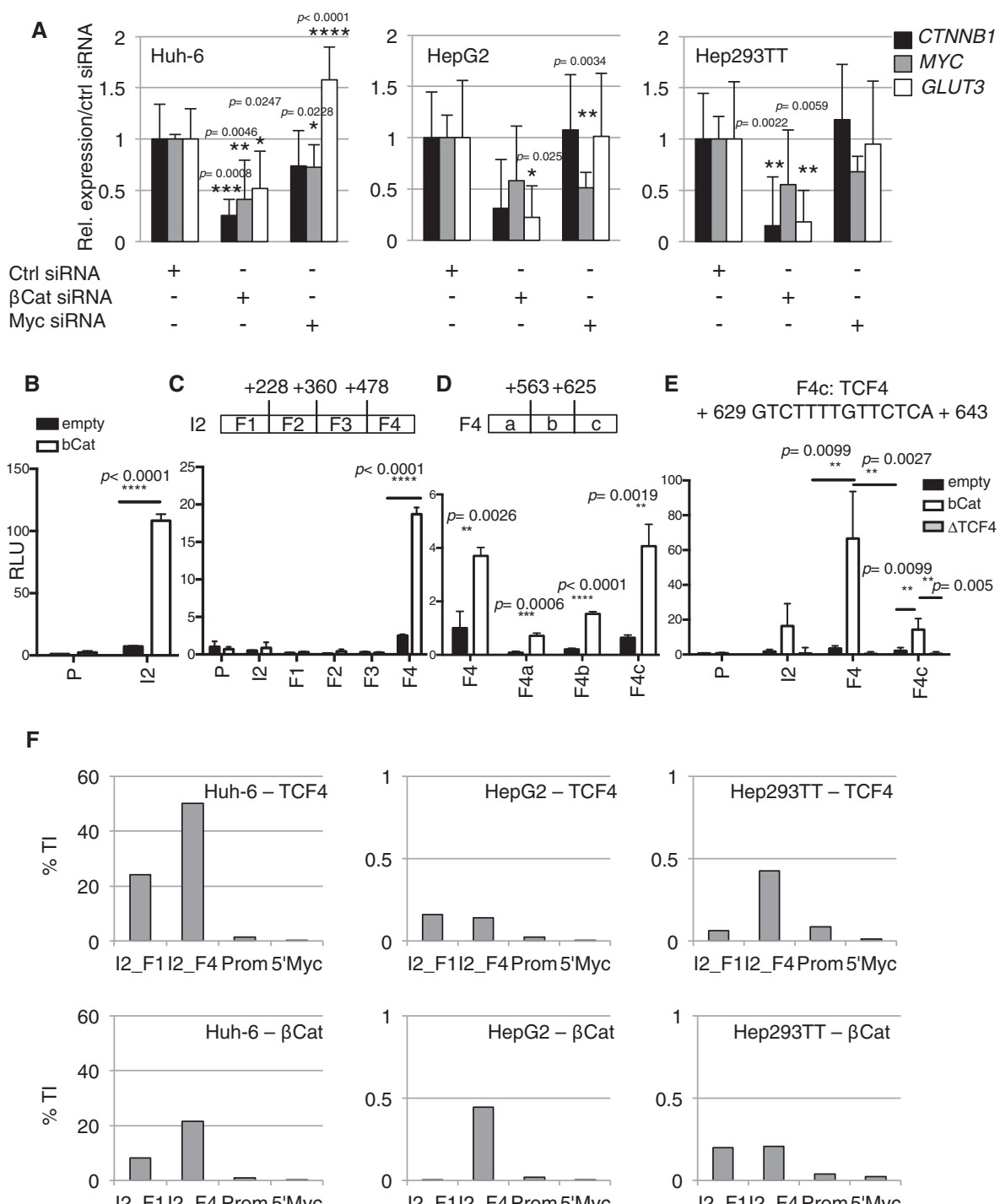

**Figure 5. *GLUT3* is a direct target of TCF4/β-catenin.**

A   Real-time PCR analysis for *CTNNB1*, *MYC*, and *GLUT3* expression in HB cell lines after siRNA transfection. Data show means ± s.d. (n = 3). P-values were determined by Mann–Whitney test.

B   The promoter region (P) and the intron 2 (I2) of *GLUT3* were cloned into a luciferase reporter and measured as relative light unit (RLU) from HEK 293T cells transfected with the luciferase reporter and β-catenin expressing vector or an empty vector as negative control.

C   Luciferase activity was measured as in (B) with *GLUT3* intron 2 separated into four different fragments.

D   Luciferase activity was measured as in (B) with *GLUT3* intron 2F4 divided into three fragments.

E   Luciferase activity was measured as in (B). For I2, F4, and F4c, either wild-type or a variant with a deletion of the TCF4 binding site (ΔTCF4) was used.

F   Chromatin extract from the indicated cell lines was used for TCF4 (upper panel) or β-catenin (lower panel) ChIP-real-time PCR. A 5-kb region upstream of the *MYC* gene (5′ Myc) was used as negative control. Experiments were performed in triplicates per each condition and analyzed separately. See Appendix Fig S3A and B.

Data information: For (B–E), data show means ± s.d. (n = 5) and P-values were determined by Mann–Whitney test.

involved in interaction with α-catenin (Fig EV3A). To prove that fetal-type β-catenin long deletion mutants have lost their ability to interact with α-catenin, we performed co-immunoprecipitation experiments. Of the four VSV-tagged β-catenin constructs (representing each mutant from the four HB cell lines), FLAG-tagged α-catenin interacted with the Huh-6 and the Hep-U2-derived mutants, but none of the HepG2 or Hep293TT ones (Fig EV3B). Additionally, in fetal-like HB cells, only wild-type endogenous β-catenin, not the fetal Δex3-4 mutants despite strong expression, bound endogenous α-catenin (Fig EV3C). For Huh-6 cells, although there was a clear interaction, it was not possible to know which of wild-type, the point mutant, or both β-catenin proteins were involved. Hence, our results demonstrate that fetal-specific HB β-catenin (Δex3-4) mutants are unable to bind α-catenin.

### Different HB subtypes are sensitive to specific glycolysis inhibitors

In order to determine whether the specific embryonal-like cell metabolism can be exploited to reduce proliferation and survival we decided to interrogate the dependency on glycolysis. We used two compounds: 2-DG and 3-bromopyruvate (3BP). 2-DG is phosphorylated by hexokinase but not metabolized further, whereas 3BP inhibits HK2 (Pedersen, 2012; Shoshan, 2012). Strikingly, apoptosis was triggered in Huh-6 cells submitted to 2-DG, which was already visible with 25% of 2-DG, as indicated by Annexin-V positivity and by an increased caspase-3/7 activity (Fig 6A and B, Appendix Fig S5A and B). In contrast, HepG2 cells did not respond to 2-DG incubation, whereas Hep293TT showed only an intermediary response. We confirmed the strongest Huh-6 response by monitoring ATP production after 2-DG treatment in the three cell lines (Appendix Fig S5C).

In marked contrast, 3BP affected fetal-like cells preferentially, as indicated by the increase in Annexin-V staining and caspase-3/7 activity observed in HepG2 and Hep293TT compared to Huh-6 (Fig 6C, Appendix Fig S5D and E) and by the reduced production of ATP in fetal compared to embryonal-like HB cells (Fig 6D). These data support the known HK2 inhibitory effects of 3BP and are consistent with more expressed *HK2* in fetal-like cells. We next tested the possibility of an HK-isoform-specific dependency for the two groups of HB. Congruent with strong *HK1* levels in Huh-6, siRNA-mediated *HK1* but not *HK2* gene silencing led to an increased cell death as revealed by Annexin-V staining. In contrast, HepG2 and Hep293TT were affected by *HK2* but not *HK1* silencing (Fig 6E and F). Altogether, our results demonstrate that embryonal-like cells differ from fetal-like cells with a distinct metabolic dependency with different sensitivity to glycolysis inhibitors and that the survival of the two HB subtypes relies on different HK isoforms.

### LDHB and G6PC stain different human HB components

In an effort to validate our metabolic subtypes *in vivo* and in order to identify new diagnostic biomarkers in addition to GLUT3, we characterized the expression of LDHB and G6PC, whose coding genes were differentially expressed in embryonal compared to fetal-like HB cell lines (see Fig 3A, Datasets EV1 and EV2). In non-tumor liver, LDHB was moderately expressed in the cytoplasm and nucleus of interlobular bile duct cells, but never in hepatocytes. As expected, G6PC showed moderate cytoplasmic staining of the non-tumor hepatocytes while cholangiocytes remained negative.

In the fetal component, tumor cells showed a diffuse reactivity to G6PC, with the exception of two tumors characterized by focal staining (Fig 7A, B and J). Reactivity ranged from mild (1+) to strong (3+), and the pattern of staining was mainly cytoplasmic. On the other hand, 16 out of 30 fetal tumors were negative for LDHB expression, with only a slight reactivity along the sinusoids (Fig 7C and K). In contrast, the vast majority of tumors displaying embryonal differentiation (21/23) expressed LDHB (Fig 7D, F and K). Of note, LDHB even revealed small foci of cells with a high nuclear cytoplasmic ratio consistent with embryonal cells, which had not been detected on standard H&E or HPS (Fig 7I). The embryonal foci of HB tumors remained negative for G6PC staining in nearly 50% of the cases (12/22) (Fig 7E and J). In one embryonal tumor positive for G6PC, cells tended to be larger and more mature appearing (Fig 7G and H).

These analyses from primary tumors confirmed our *in vitro* findings and collectively reveal that embryonal and fetal HBs, which represent two distinct histological components of HB, could be metabolically classified based on the expression of LDHB and G6PC, markers of glycolysis and gluconeogenesis. Based on our results, GLUT3 as well as LDHB and G6PC staining could serve as novel diagnostic tools in addition to histological classification to guide HB diagnosis, highlighting in some instances minor cell components not readily identified on routine standard stains.

## Discussion

In this study, we performed a transcriptome profile to investigate the molecular differences between adult HCC and pediatric HB, which led us to identify an enhanced sugar transport and usage in HB tumors. Moreover, we were able to identify two different metabolic subtypes of HB based on the type of glucose metabolism, with the embryonal-like cells displaying increased glycolysis and the fetal-like expressing stronger levels of gluconeogenesis genes. We also described a novel role for β-catenin in the regulation of glucose transporter GLUT3 expression. This allowed the discovery of three new robust biomarkers to distinguish between the different HB subtypes.

Although a majority of tumors harbor mutations in *CTNNB1*, long deletions encompassing exons 3 and 4 are only detected in pure fetal HB, and correlate with increased Notch versus Wnt pathway activation (Lopez-Terrada *et al*, 2009). Conversely, embryonal HBs express small deletions or missense mutations in *CTNNB1* and show Wnt over Notch activation. These findings are compatible with tumorigenesis taking place in the context of early differentiation stage during liver development for embryonal HB, and later stage for fetal HB. Our identification of increased expression of gluconeogenesis-related genes in fetal-like HB cells also reflects their increased differentiation, closer to mature hepatocytes. By extension, this intricate connection between organogenesis and tumorigenesis is particularly relevant for pediatric solid malignancies, where tumorigenesis arises within developmentally immature tissue environments (Scotting *et al*, 2005). A recent gene expression profiling has enabled a molecular classification of HB, revealing two groups

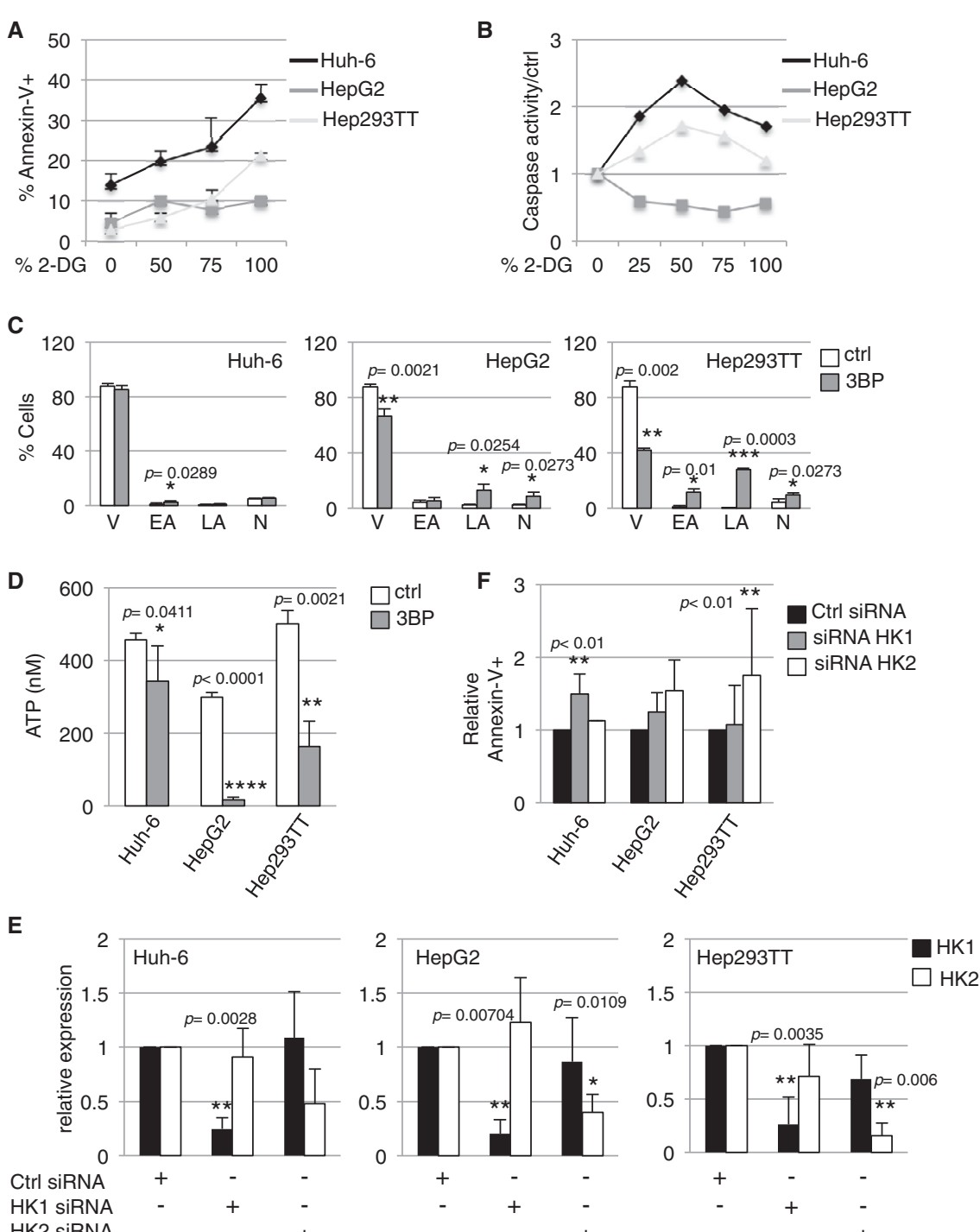

**Figure 6. The two groups of HB cell lines are sensitive to different inhibitors of glycolysis and use different hexokinases.**

A  Analysis of Annexin V staining 24 h after 2-DG treatment. The indicated cell lines were stained with Annexin V–APC antibody. Data show means ± s.d. (n = 3).

B  Analysis of caspase-3/7 activity in the indicated cell lines cultured in presence of 2-DG compared to control. Experiments were performed in triplicates per each condition and analyzed separately. Data show one representative replicate out of three independent experiments (see Appendix Fig S5A and B for the other replicates).

C  The indicated cell lines were stained with Annexin V 12 h after treatment with 3BP to estimate the percentage of viable (V) early (EA) late (LA) apoptotic, and necrotic (N) cells. Data show means ± s.d. (n = 3).

D  ATP production was measured in the indicated cell lines 12 h after 3BP treatment. Data show means ± s.d. (n = 3).

E  The indicated cell lines were transfected with *HK1*, *HK2*, or control siRNA. *HK1* and *HK2* expressions were analyzed by real-time PCR. Data show means ± s.d. (n = 3).

F  The indicated cell lines were stained with Annexin V–APC antibody 72 h after transfection with *HK1* siRNA, *HK2* siRNA, or control siRNA. Data show means ± s.d. (n = 4).

Data information: *P*-values were determined by Mann–Whitney test. For (A–D), normal medium was used as control.

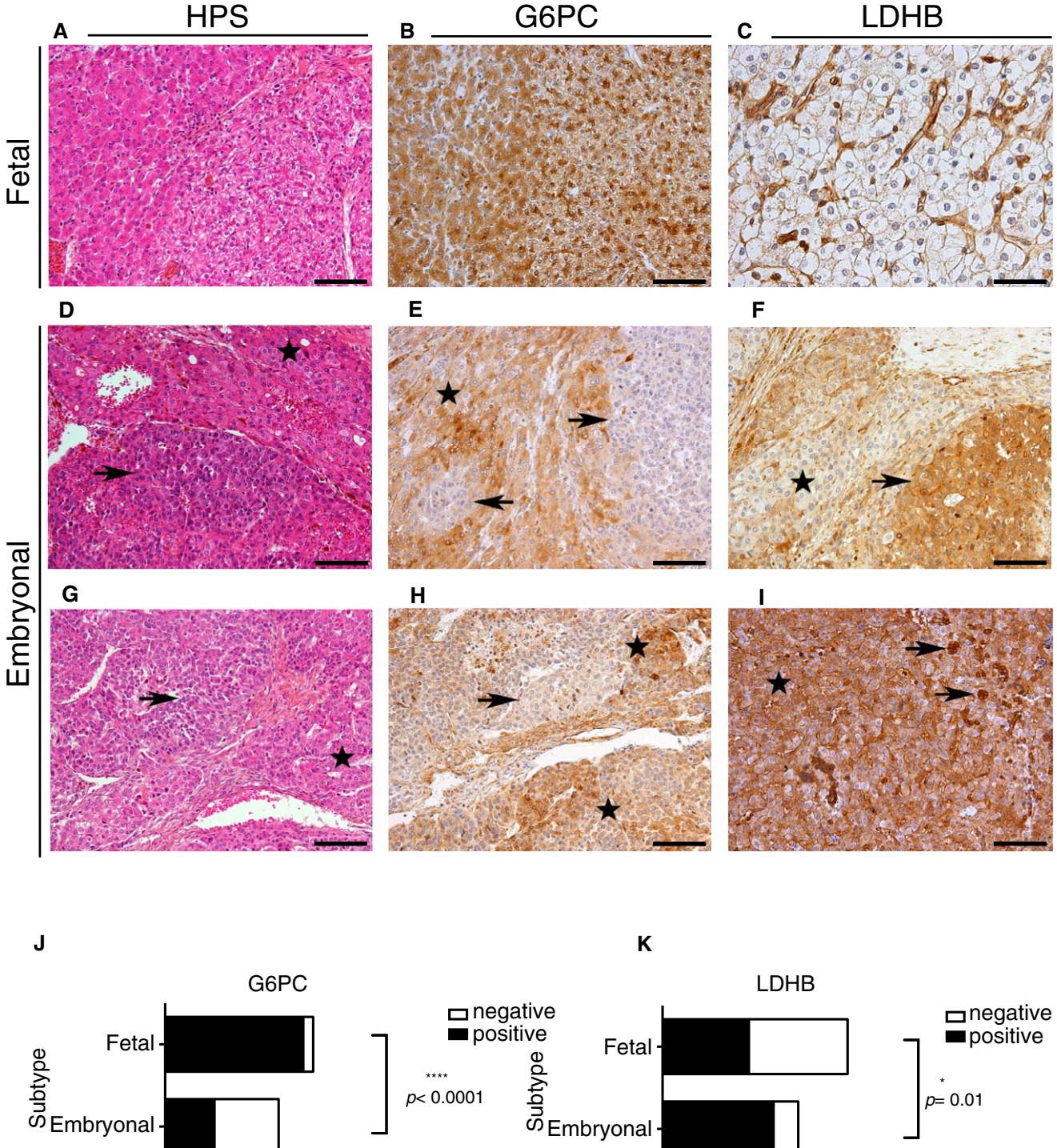

**Figure 7.   LDHB and G6PC stain different components of human HB tumors.**

A–C  Fetal component. On the right-hand side, irregular distribution of glycogen imparts a "light" pattern on HPS (A), and perinuclear "clumped" G6PC accumulation (B). The fetal tumor cells do not react to LDHB (C). Instead, LDHB reactivity is observed along the sinusoids.

D–I  Embryonal component. Tumors displaying embryonal differentiation expressed LDHB (F). LDHB further highlights small foci of cells with a high nuclear cytoplasmic ratio consistent with embryonal cells (I). The embryonal tumor cells remained negative for G6PC (E). Only one tumor (G) with embryonal cells that tended to appear larger and more mature showed focal G6PC reactivity (H). Arrows: embryonal component; stars: fetal component. Scale bars: 100 μm (original magnification ×200), except for (C), scale bar: 50 μm (original magnification ×400).

J, K  Contingency table of G6PC and LDHB staining depending on the embryonal/fetal status of the tumor. Chi-square test was used.

evoking distinct phases of liver development. One of them, called C2, was enriched for embryonal tumors, expressed hepatic stem/progenitor markers of immature liver, and was associated with bad prognosis (Cairo *et al*, 2008). Interestingly, gene expression from C2 tumors clustered with that of early liver development in the mouse (E11.5 and 12.5). *HK1*, which we found overexpressed in embryonal compared to fetal-like tumor cells, is abundant in the developing liver, with levels dramatically reduced at birth. In contrast, *HK2*, whose expression is low in developing and adult liver, is the most commonly upregulated hexokinase isoform in adult tumors. This reveals alternative strategies in hexokinase utilization, adult tumors and fetal HB expressing mostly HK2, embryonal HBs relying on the embryonic isoform, HK1. This may open therapeutic opportunities, as this knowledge could enable the future development of isoform-specific hexokinase small molecule inhibitory compounds. Similarly, our study reveals that a possible target of aggressive HB would be GLUT3 itself. The development of molecules transported into tumor cells through GLUT3 to deliver toxic compounds or to block glucose transport activity might be facilitated by the recent X-ray crystal structure determination of this transporter and other members of this family, GLUT1 and GLUT5 (Deng *et al*, 2014, 2015; Nomura *et al*, 2015).

The modulation of *GLUT3* levels in cancer appears to depend on the activity of various factors. Indeed, in tumor cells from colorectal cancer, *GLUT3* was induced by caveolin-1, in a HMGA1-dependent manner, HMGA1 enhancing *GLUT3* transcription by binding to specific sites within its promoter region (Ha *et al*, 2012). In cells derived from NSCLC or HCC, *GLUT3* was induced in response to ZEB1 activity during EMT (Masin *et al*, 2014), and *GLUT3* is known to respond to NF-κB (Kawauchi *et al*, 2008). In this study, we found that *GLUT3* is a direct target gene of TCF4/β-catenin in tumor cells from HB. TCF4/β-catenin binding in the *GLUT3* gene occurs at the end of intron 2, in the same intron, and only ~250 bp after the ZEB1 binding site (Masin *et al*, 2014), demonstrating a central regulatory role for this intragenic enhancer in cancer cells.

We showed that mutant β-catenin interacts with α-catenin exclusively in HB cells with alterations confined to *CTNNB1* exon 3, larger deletions spanning exon 3 to 4 found in fetal HB tumors, HepG2 and Hep293TT preventing binding. α-Catenin plays a critical role in cell–cell junction (Nelson & Nusse, 2004) and in regulation of gene transcription (Choi *et al*, 2013; Daugherty *et al*, 2014; McCrea & Gottardi, 2016). Considering this, it will be important to delineate the contribution of the β-catenin/α-catenin complex in fine-tuning the Wnt-dependent transcriptional program and/or metabolic reprogramming in embryonal versus fetal HB tumors.

A limitation of our study is the lack of animal models. To study HB, a limited number of genetically engineered mouse models exist, in which one of *Myc*, *LIN28B*, or *Ctnnb1* oncogenes can be induced in the mouse liver (Shachaf *et al*, 2004; Mokkapati *et al*, 2014; Nguyen *et al*, 2014) leading to HB or HCC development. Based on our findings, we suggest concomitant oncogene activation and metabolic pathway alteration will enable to refine these models to generate mice developing histologically defined subtypes of HB. The elaboration of improved models could be useful for preclinical trials aiming to identify better treatment strategies against human HB.

The heterogeneity of HB tumors has enabled to classify them first based on histological criteria, followed by molecular signatures. In our study, we demonstrated that different subtypes of HB differ metabolically. Indeed, embryonal but not fetal HB tissues express GLUT3 and LDHB proteins, while fetal HBs express G6PC preferentially. Although they might not be readily usable in clinical settings alone, these new potential biomarkers might help discriminate between different tumor subtypes as a complement to conventional histopathology. We hope that future investigations will enable the discovery of additional metabolic markers, whose distinct expression in different components of HB could help for clinical diagnosis. Finally, our study may open an avenue for new strategies, based on metabolic vulnerabilities, to treat aggressive HB.

# Materials and Methods

### Cell culture conditions

HB and HCC cell lines were kindly provided by J. Gouttenoire and D. Moradpour (University of Lausanne). HB cell lines Hep-U2 and Hep293TT were from Dr. S. Brüderlein (University of Ulm, Germany) (Scheil *et al*, 2003) and Prof. GE Tomlinson (University of Texas, San Antonio, TX) (Chen *et al*, 2009). Cells were cultured in RPMI 1640 medium (ThermoFisher) with L-glutamine, supplemented with 10% heat-inactivated fetal bovine serum, 1% penicillin/streptomycin, 1 mM sodium pyruvate (ThermoFisher). The addition of 25 mM HEPES buffer was required for Hep293TT cell culture.

### ChIP-PCR

Chromatin was prepared from $10^7$ HB cells crosslinked with 4% PFA as previously described (Rowe *et al*, 2013). Chromatin was fragmented using a Covaris sonicator and immunoprecipitated using the following antibodies: anti-TCF4 (Millipore, #05-511) or anti-β-catenin (Abcam, ab32572). DNA was reverse-crosslinked O/N at 65°C by adding proteinase K (Promega) and RNAse A (Sigma-Aldrich), purified using the Mini Elute PCR Purification Kit (Qiagen), and subjected to real-time PCR. Enrichment was calculated as total input percentage (% TI). The following primers were used: F4_for ACATCGGTGCTGCCACCTAC, F4_rev GGTTGGTGGAAGAA CAGAC; F4c_for GGAAGGAAATGATCCCTAAT, F4c_rev GGTTG GTGGAAGAACAGAC; Prom_for GGGATTACAAGTGTGAGCCACC, Prom_rev TGAAGAATCACCAGCTTCTTGG; 5′ Myc_for GCCTCACA AAGTGCTAGGATTA, 5′ Myc_rev CGGCCTCACAGAACAGAATAG.

### Glucose uptake measurement

Briefly, cells were seeded the day before the experiment. All buffers, except the one used for lysis, were freshly prepared. Cells were washed 3× with 37°C wash buffer (2 mM pyruvic acid in Krebs–Henseleit buffer: 1.18 mM $KH_2PO_4$, 1.18 mM $MgSO_4.7H_2O$, 2.5 mM $CaCl_2.2H_2O$, 20 mM HEPES, 137 mM NaCl, 4.7 mM KCl) and exposed to 37°C background buffer (50 mM D-glucose, 100 μM 2-DG, 1 μCi/ml 2-[2,6 $^3$H]DG in PBS) or 37°C transport buffer (100 μM 2-DG, 1 μCi/ml 2-[2,6 $^3$H]DG in wash buffer). 10 min after

start of exposure, transport was arrested with addition of stop buffer (50 mM D-glucose in PBS 1×). Three washes using stop buffer were made prior to lysis 100 mM NaOH, 0.1% SDS). Technical duplicate samples were made, in addition to two samples used for normalizing containing 30 μl of either background or transport buffer completed with 170 μl of lysis buffer. Following addition scintillation liquid to each sample, radioactivity was determined by scintillation counting. For sample normalization, protein concentration was measured by BCA protein assay. Specific transport was calculated by subtracting background well values to the corresponding transport values.

### Seahorse analysis

OCR and ECAR measurements were performed using the XF24 or XF96 Extracellular Flux analyzer (Seahorse Bioscience, North Billerica, MA). Briefly, cells were plated into XF96 (V3) polystyrene cell culture plates (Seahorse Bioscience, North Billerica). The cells were incubated for 24–28 h in a humidified 37°C incubator with 5% $CO_2$ (RMPI1640 medium). Prior to assay, cells were incubated in a 37°C/non-$CO_2$ incubator for 60 min. All experiments were performed at 37°C. Each measurement cycle consisted of a mixing time of 3 min and a data acquisition period of 2 min. Each measurement was taken in triplicates. For glycolysis assay, 10 mM glucose was added during seahorse experiment followed by 50 mM 2-DG after 24 h in RPMI medium without glucose complemented with 5% serum. Palmitate–BSA consumption was performed using the Seahorse XF Palmitate–BSA FAO Substrate (Agilent) using the manufacturer's instructions. OCR and ECAR data points refer to the average rates during the measurement cycles. Because the cell lines proliferate at different rates during the 24- to 28-h incubation period, proteins were extracted using RIPA buffer to determine protein concentration in each well after an assay. OCR and ECAR were reported as normalized against protein concentration.

### Statistics

Unless specified differently, *P*-values were determined by Mann–Whitney tests.

### Data availability

The RNA sequencing data from this publication have been deposited to the GEO database (https://www.ncbi.nlm.nih.gov/geo/) and assigned the identifier: GSE83518.

**Expanded View** for this article is available online.

### Acknowledgements

We thank Lorenzo Petrini, Caroline Contat, Silvia Sabatino, Hongbo Zhang, and the EPFL SV Histology and Flow Cytometry Core Facilities for technical assistance. We thank the Lausanne Genomic Technologies Facility for RNA sequencing. We thank the Protein Modeling Facility of the University of Lausanne for the support in Molecular Modeling and the Vital-IT group of the SIB Swiss Institute of Bioinformatics for providing computational resources. We thank Jérôme Gouttenoire and Darius Moradpour (CHUV, Lausanne) for critical reading of the manuscript and Barbara Wildhaber (HUG, Geneva and CHUV, Lausanne) for helpful discussions. This work was supported by the Swiss National Science Foundation (PP00P3_133661 and PP00P3_157527) and by FORCE, a Foundation for Children Cancer Research.

### The paper explained

#### Problem

Hepatoblastoma (HB) is the most common form of pediatric liver cancer. It is characterized by frequent gain-of-function mutations in *CTNNB1* encoding β-catenin. Currently, it is unknown whether the molecular specificities or the histological heterogeneity in HB could be linked to a variability in tumor cell metabolism. Such knowledge could help for the development of future therapies that aim to target deregulated cellular energetics of tumors.

#### Results

Here, we provide evidence for differences in glucose usage by tumor cells from HB compared to hepatocellular carcinoma. We demonstrate that the gene encoding glucose transporter GLUT3 is a direct target of TCF4/β-catenin, and that, in tissue sections, GLUT3 protein stains embryonal but not fetal components of HB. *In vitro*, embryonal-like cells show enhanced glycolysis compared to fetal-like cells and differ by their dependency on hexokinase 1 instead of hexokinase 2 isoform for survival. We also identified two metabolic genes, *LDHB* and *G6PC*, whose expression is elevated in embryonal or fetal-like cells, respectively. In tumor tissue sections, LDHB and G6PC staining revealed their utility as new biomarkers to discriminate embryonal from fetal components of tumors.

#### Impact

Our study defining *GLUT3* as a Wnt/β-catenin target gene may have implications for metabolic reprogramming and the development of GLUT3 inhibitory compounds in HB and other Wnt-driven malignancies. The dependency of embryonal-like HB tumor cells on hexokinase 1, as well as the GLUT3, LDHB, and G6PC staining on tumor sections could be used as novel biomarkers to highlight metabolic variation in HB and guide future therapies based on metabolic vulnerabilities.

### Author contributions

SC and P-BA performed most of the experiments; JV and EM performed some experiments; PA and MD did the RNAseq analyses; CG and A-LR did the histopathology analysis from HB tumor sections; VZ and OM did the molecular modeling; SC and EM conceptualized the project; SC, P-BA, and EM analyzed the data; EM supervised the project; SC, P-BA, and EM wrote the manuscript.

### Conflict of interest

The authors declare that they have no conflict of interest.

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
