## [Review Process File · EMBO Molecular Medicine]

Mutant CTNNB1 and histological heterogeneity define metabolic subtypes of hepatoblastoma

Stefania Crippa, Pierre-Benoit Ancy, Jessica Vazquez, Paolo Angelino, Anne-Laure Rougemont, Catherine Guettier, Vincent Zoete, Mauro Delorenzi, Olivier Michielin, Etienne Meylan

Corresponding author: Etienne Meylan, Ecole Polytechnique Fédérale de Lausanne

Review timeline:

Submission date:	17 March 2017
Editorial Decision:	12 April 2017
Revision received:	13 July 2017
Editorial Decision:	14 August 2017
Revision received:	16 August 2017
Accepted:	18 August 2017

Transaction Report:

Editor: Roberto Buccione

1st Editorial Decision

12 April 2017

Thank you for the submission of your manuscript to EMBO Molecular Medicine. We have now heard back from the Reviewers whom we asked to evaluate your manuscript.

As you will see, reviewers 1 and 3 provide detailed evaluations, which highlight the interest of the manuscript but also point to important and partially overlapping issues including the need for further experimentation to consolidate and fully support some of the main claims. Reviewer 2, although quite cursory, is perhaps more reserved. S/he also notes, in agreement with reviewer 3, that myc knockdown was not effective, which of course somewhat jeopardises the conclusions based on these experiments. Reviewer 2 also requests further in vitro and in vivo experimentation concerning based on Glut3 GOF and LOF.

After further discussion, it was agreed that you should be allowed to submit a revised manuscript, with the understanding that the reviewers' concerns must be addressed in full with additional experimental data where appropriate and that acceptance of the manuscript will entail a second round of review. However, I am willing to forego the further in vivo experimentation mentioned by reviewer 2, provided you discuss the point in your rebuttal.

It is important that you consider that it is EMBO Molecular Medicine policy to allow a single round of revision only and that, therefore, acceptance or rejection of the manuscript will depend on the completeness of your responses included in the next, final version of the manuscript.

As you know, EMBO Molecular Medicine has a "scooping protection" policy, whereby similar findings that are published by others during review or revision are not a criterion for rejection.

However, I do ask you to get in touch with us after three months if you have not completed your revision, to update us on the status. Please also contact us as soon as possible if similar work is published elsewhere.

Finally, please note that EMBO Molecular Medicine now requires a complete author checklist (<http://embomolmed.embopress.org/authorguide#editorial3>) to be submitted with all revised manuscripts. Provision of the author checklist is mandatory at revision stage; The checklist is designed to enhance and standardize reporting of key information in research papers and to support reanalysis and repetition of experiments by the community. The list covers key information for figure panels and captions and focuses on statistics, the reporting of reagents, animal models and human subject-derived data, as well as guidance to optimise data accessibility.

Please note that we now mandate that all corresponding authors list an ORCID digital identifier. You may do so through our web platform upon submission and the procedure takes <90 seconds to complete. We also encourage co-authors to supply an ORCID identifier, which will be linked to their name for unambiguous name identification.

Last, but not least, please carefully conform to our author guidelines (<http://embomolmed.embopress.org/authorguide>) to ensure rapid pre-acceptance processing in case of a favourable outcome on your revision.

I look forward to seeing a revised form of your manuscript in due time.

***** Reviewer's comments *****

Referee #1 (Remarks):

The global aim of this study is to identify specific markers and putative targets for the distinct types of hepatoblastoma, which is the most common malignant tumor of the liver in children. The authors show that embryonal HB differentially expresses metabolic genes, compared to fetal HB. This correlates with the aggressivity of the tumors, being pure fetal tumors of better prognostic than embryonic tumors. It is interesting the finding that genes involved in the glycolysis or gluconeogenesis pathways are amongst the differentially expressed mRNAs and protein, suggesting the importance of this pathway in the etiology of this cancer. A particular effort has been directed to study the expression and regulation of the glucose transporter GLUT3, which is specifically up-regulated in embryonal HB. This correlates with increased glycolysis in these tumor cell lines and with higher activity of the Wnt pathway. Indeed the authors show that GLUT3 expression is under the control of β -catenin. The expression of these metabolic genes is also analyzed in human samples. This is a well-conducted study, and the results may be readily used in the clinical setting. Metabolic differences between fetal and embryonic origin HB are very relevant, not only as a diagnostic marker but also for the understanding the complex relationship between metabolism and cancer. There are, however some issues that would further increase the quality of the paper.

1. Some experiments are poorly described. For instance, what is the basis of the experiment shown in figure 2F about the oxidative capacity of mitochondria? Please better explain.
2. In figure 2D it is shown that glucose transport is very high in HuH6 cells, compared to fetal HB cells (HepG2 and Hep293TT) and mesenchymal cells (HLE, HLF). However, these last cells show a high rate of glycolysis in the Seahorse experiment (figure 2E). How is this possible? Similarly, HepG2 cells show a high oxidative capacity. Where is the glucose coming from?. May be a better way to demonstrate glycolytic capacity is starving cells and adding glucose to the cell media in the Seahorse, and then measure ECAR and OCR.
3. Another issue regarding the metabolic changes that take place in these cells is fatty acid oxidation. Seahorse analyses could be very informative in this regards. Showing that the decreased glucose use is correlated with increased FA use would be very relevant to illustrate the metabolic switch.
4. Aerobic glycolysis is typically correlated with increased biosynthesis, notably de novo fatty acid

synthesis. Analyses of this process, for instance by quantifying gene expression would be appreciated.

5. This reviewer does not understand why different methods are tested to measure the effects of 2Dg and 3BP treatments.

6. The expression of HK1 is high in HuH6 cells, whereas HK2 seems more specific for HepG2 cells. It is surprising that the authors do not test the expression of these proteins in the human HB samples. Can the expression of HKs discriminate embryonal from fetal HB?

Referee #2 (Comments on Novelty/Model System):

Lack of in vivo animal cancer models

Referee #2 (Remarks):

The manuscript by the Meylan group compares the gene expression of different hepatoblastoma (HB) and hepatocellular carcinoma (HCC) cell lines. They validate some of the differentially expressed gene products in human biopsies: Glut3, G6PC, LDHB. Finally, they give some hints on the transcriptional regulation of Glut3 by beta catenin.

Although some of the data are interesting and convincing, the manuscript lacks focus and in depth demonstration.

- 1) The relevance of Glut3 expression requires functional in vitro and in vivo studies using loss- and gain-of-function approaches
- 2) The myc siRNA approach does not seem effective, as substantial residual myc expression can be observed (fig. 5A)
- 3) Glut3, G6PC and LDHB do not appear to reliably discriminate between tumor types as 30-50% of samples are not consistent with the rest (fig. 4C, 7I, 7J).

Referee #3 (Remarks):

In the present study Crippa and colleagues provide an extension of the definition of hepatoblastoma (HB) subtypes, representing the most common liver cancer in children. Mainly based comparisons between cell lines representing previously established embryonal and fetal HB subtypes, as well as different hepatocellular carcinoma cell lines, representing the most common form of liver cancer in adults, the authors describe molecular and metabolic differences contributing to subtype definition. Specifically, the authors provided data suggesting a more glycolytic metabolic phenotype characteristic for the embryonal HB subtype, which was mainly attributed to TCF4/beta-Catenin-dependent induction of glucose transporter (GLUT) 3 expression. In addition, GLUT3 and other metabolic proteins were proposed as biomarkers for diagnosis of subtypes, also potentially offering novel treatment opportunities based on metabolic vulnerabilities.

The presented insights into mechanisms defining subtypes based on oncogene-dependent metabolic reprogramming are interesting and of potential translational value. Although most of the provided data are derived from in vitro experiments, some of the central findings concerning the metabolic markers have been confirmed in human HB samples. However, some of the data presented fall short of the characterization of the metabolic phenotype. Also, some of the experimental results were confusing and some of the figures did not have clear labeling/figure legends, making it difficult to interpret the data. Therefore, the study at this stage requires revision addressing the following points:

Major Points:

Fig. 2E: It is surprising that while that the authors provided Seahorse data, they seemingly did not exploit the full potential of the method concerning the metabolic phenotyping of the cells. The authors should measure extracellular acidification in the course of the so-called "glycolysis stress test", which includes sequential addition of glucose, oligomycin (assessing maximum glycolytic capacity) and 2-DG (inhibition of glycolysis), drawing a much clearer picture of the glycolytic phenotype in comparison to the cell line comparisons under basal conditions only. For selected cell lines (e.g. Huh-6 vs Hep293TT) Seahorse glycolysis assays (as described above) or alternatively glucose uptake assays should be performed in the presence and absence of Glut3 and/or

beta-catenin knockdown, thereby providing more robust evidence for the central role of beta-catenin and/or Glut3 for the "sugar metabolic phenotype" of the embryonal-like subtype of HB.

Minor Points:

Fig. 2A: It is unclear from which material the presented RNAseq data were derived from. Does HB or HCC refer to a specific cell line of the respective type or do the gene expression levels represent mean values of several cell lines? Please specify accordingly

While the Supplemental tables as excel sheets are informative, the provided PDF formats are not useful and should be deleted.

Fig5A; page9 line 15-17: Myc expression is more strongly reduced upon knockdown of beta-catenin than in the samples with direct Myc KD, which was obviously not very efficient. This is questioning whether these data really allow the conclusion that Myc does not exert any effect on Glut3. Also, why did the authors not include the corresponding Myc knockdown samples in the corresponding analysis shown in Fig. S4A. Please comment.

Fig.6, page 12; discussion page 15: How do elevated levels of HK2 (also reflected at the level of higher sensitivity to 3BP) relate to reduced glycolytic activity in HB-fetal lines vs elevated glycolysis in embryonal-like HK1-induced cells? Please add a comment to the functional consequences of different HK isoforms in the different subtypes with respect to the glycolytic phenotype.

Information related to HB surgical specimens is missing in the Material and Methods section.

1st Revision - authors' response

13 July 2017

Referee #1 (Remarks):

The global aim of this study is to identify specific markers and putative targets for the distinct types of hepatoblastoma, which is the most common malignant tumor of the liver in children. The authors show that embryonal HB differentially expresses metabolic genes, compared to fetal HB. This correlates with the aggressivity of the tumors, being pure fetal tumors of better prognostic than embryonic tumors. It is interesting the finding that genes involved in the glycolysis or gluconeogenesis pathways are amongst the differentially expressed mRNAs and protein, suggesting the importance of this pathway in the etiology of this cancer. A particular effort has been directed to study the expression and regulation of the glucose transporter GLUT3, which is specifically up-regulated in embryonal HB. This correlates with increased glycolysis in these tumor cell lines and with higher activity of the Wnt pathway. Indeed the authors show that GLUT3 expression is under the control of β -catenin. The expression of these metabolic genes is also analyzed in human samples. This is a well-conducted study, and the results may be readily used in the clinical setting. Metabolic differences between fetal and embryonic origin HB are very relevant, not only as a diagnostic marker but also for the understanding the complex relationship between metabolism and cancer. There are, however some issues that would further increase the quality of the paper.

Some experiments are poorly described. For instance, what is the basis of the experiment shown in figure 2F about the oxidative capacity of mitochondria? Please better explain.

First, we would like to thank Reviewer #1 for the insightful comments.

To label mitochondria, cells were incubated with MitoTracker® probes, which passively diffuse across the mitochondrial membranes and accumulate in active mitochondria. In our experiment we used two different mitotracker probes: the MitoTracker® Green probe allows the detection of the entire mitochondrial mass, used in the experiment to normalize for the mitochondrial content. In addition, we also used a rosamine-based probe, MitoTracker® Red. This reduced probe does not fluoresce until it gets oxidized, reflecting the oxidation ability of the mitochondria. This experiment is now better explained in the Appendix Supplementary Text, page 7.

In figure 2D it is shown that glucose transport is very high in HuH6 cells, compared to fetal HB cells (HepG2 and Hep293TT) and mesenchymal cells (HLE, HLF). However, these last cells show a high rate of glycolysis in the Seahorse experiment (figure 2E). How is this possible? Similarly, HepG2 cells show a high oxidative capacity. Where is the glucose coming from?. May be a better way to demonstrate glycolytic capacity is starving cells and adding glucose to the cell media in the Seahorse, and then measure ECAR and OCR.

In our experiments, the glucose is coming from the RPMI medium and from the serum. We agree with this reviewer that addition of glucose during the seahorse after a 24 hours glucose starvation can give a clearer picture of the glycolytic potential of the cells. Following the reviewer's recommendations, we performed this experiment. As a result, we found that Huh-6 exhibited the highest glucose response in comparison to all other cell lines (Figure 1 below). Interestingly, we also observed a response to glucose in Hep293TT confirming the glycolytic potential of these cells that we previously found with MitoTracker. We did not observe a clear response to glucose in the other tested cell lines. We added this result as new Figure 2F in the revised version of the manuscript.

Figure 1: After 24 hours starvation, 10 mM glucose were added to the well followed by 2-DG to block glycolysis at a concentration of 50 mM. Sequential measurements of ECAR and after the different injections were performed.

Another issue regarding the metabolic changes that take place in these cells is fatty acid oxidation. Seahorse analyses could be very informative in this regards. Showing that the decreased glucose use in correlated with increased FA use would be very relevant to illustrate the metabolic switch.

We thank the reviewer for this suggestion about fatty acid oxidation that was not considered in the previous version of the manuscript. As it was suggested, we measured OCR after addition of palmitate-BSA or BSA alone (vehicle control) after 24 hours in limited medium (no glucose, 1% FBS). As a result, only HepG2 were consuming palmitate while we did not observe a response for Huh-6 (Figure 2A below). A trend for Hep293TT was also observed, although it did not reach significance. Hence, decreased glucose use indeed correlates with increased FA use. To understand better the molecular basis for fatty acid consumption in the different cell lines we took advantage of our RNAseq data to monitor the expression of CPT1A (Carnitine Palmitoyltransferase 1A), responsible for exogenous fatty acid incorporation into mitochondria. This gene was expressed in fetal-like cells with HepG2 displaying the strongest expression (Figure 2B below), but almost not expressed in embryonal-like Huh-6. In order to explore more precisely fatty acid metabolism in the different HB cell subtypes we used Etomoxir, a CPT1A inhibitor. Cells were treated with Etomoxir at a concentration of 10 mM to block exogenous fatty acid consumption. Only HepG2 oxygen consumption was affected by Etomoxir, revealing the highest exogenous fatty acid dependence in this cell type (Figure 2C below). The Figure 2A and 2C below are present in the new version of the manuscript as new Figure 2G and new Figure 2H, respectively.

Figure 2: (A) After 24 hours in limited medium (1%FBS and no glucose) BSA-Palmitate or BSA was added just before seahorse experiment. OCR was then measured. (B) RPKM of CPT1 in Huh-6, HepG2 and Hep293TT. (C) After 24 hours in limited medium (1%FBS and no glucose) BSA-Palmitate or BSA was added just before seahorse experiment in the presence or not of 10mM Etomoxir.

Aerobic glycolysis is typically correlated with increased biosynthesis, notably de novo fatty acid synthesis. Analyses of this process, for instance by quantifying gene expression would be appreciated.

Following the reviewer's comment, we monitored the expression of genes involved of *de novo* fatty acid synthesis such as FASN using our RNA sequencing data. None of the genes tested showed a significant difference in expression between embryonal-like [Huh-6, Hep-U2] and fetal-like [HepG2, Hep293TT]. We show the RPKM of some specific genes (Figure 3 below).

Figure 3: RPKM of genes involved in de novo fatty acid biosynthesis.

Additionally, to provide more information to the reader we added a supplementary table showing all the differentially expressed genes between embryonal-like cells and fetal-like cells (New Table EV2) in the revised version of the manuscript. In line with our new data on fatty acid oxidation, we performed pathway enrichment analysis (KEGG) on the up-regulated genes in fetal-like cells. We found two significant pathways in relation with fatty acid catabolism: “Fatty acid degradation pathway” (p -value= $1.50E-03$) and “PPAR signaling pathway” (p -value= $9.10E-03$). Some genes involved in those two pathways are shown below (Figure 4 below).

Figure 4: RPKM values of a panel of genes involved in fatty acid degradation that are up-regulated in fetal cells.

This reviewer does not understand why different methods are tested to measure the effects of 2Dg and 3BP treatments.

We first tried to show the effect of the different drugs by using various methods to reliably validate the metabolic distinction we observed. However, we agree with the reviewer that both treatments should be challenged with the same panel of methods to assess the cell viability. For this reason, we performed ATP production (Figure 5A below) after 2-DG treatment, and Caspase 3/7 activity after 3-BP treatment (Figure 5B below) in all hepatoblastoma cell lines. Those results confirmed the highest mortality of Huh-6 in response to 2-DG even if all cell lines tested were sensitive to 2-DG. On the other hand, when treated with the inhibitor of HK2 (3-BP) only fetal-like cells exhibited a higher Caspase activity while Huh-6 did not respond to this treatment. These two panels are now present in the Appendix Figure S5D and S5E respectively.

Figure 5: (A) ATP production quantification after 2-DG treatment in hepatoblastoma cell lines. (B) Caspase 3/7 activity after 3 BP treatment in hepatoblastoma cell lines.

6. The expression of HK1 is high in HuH6 cells, whereas HK2 seems more specific for HepG2 cells. It is surprising that the authors do not test the expression of these proteins in the human HB samples. Can the expression of HKs discriminate embryonal from fetal HB?

We thank the reviewer for this comment; in the previous version of the paper, we only considered GLUT3, G6PC and LDHB as biomarkers. According to our data, it seems that HK isoforms are good candidates to discriminate embryonal and fetal lesions. However, we had some difficulties to obtain a reliable staining. In an effort to provide another biomarker we optimized this staining and monitored the expression of HK isoforms in tumor tissues. Below is our pathology report, specifically shown in this Response Letter:

HK1 showed mild to moderate cytoplasmic staining in all fetal components assessed (16/16, Figure 6A below). Focal HK1 reactivity of variable intensity was seen in only some of the embryonal components (6/11, Figure 6B below). HK1 cytoplasmic reactivity was also seen in the squamous foci (2/2), and in the mesenchymal cells embedded in an osteoid matrix (3/3), but never in immature mesenchymal cells (2/2). We felt however that the HK1 pattern of staining was too heterogeneous, and the differences in HK1 expression between the fetal and embryonal components too subtle to consider this marker as suitable at this stage for clinical purposes.

Only a subset of the fetal tumor cells in a minority of HBs with a fetal component showed mild (1+) cytoplasmic reactivity to HK2 (2/15), whereas HK2 remained negative in all the assessed embryonal components (9/9). Squamous cells showed mild to moderate reactivity (1+/2+, 2/2).

Figure 6: A) Cytoplasmic reactivity, occasionally granular, is seen in fetal HB cells. B. The faint cytoplasmic HK1 expression is observed in embryonal cells, but not in the surrounding immature mesenchymal spindle cells. HK1, original magnification x400.

Case	HK1								HK2							
	Embryonal	Fetal	SCU	Macrotrabec	HCC-like	Squamous	Cholangiobl	IMC/Osteoid	Embryonal	Fetal	SCU	Macrotrabec	HCC-like	Squamous	Cholangiobl	IMC/Osteoid
1	Neg	1+ C granular	NA	NA	NA	NA	Neg	Neg	Neg	Neg	NA	NA	NA	NA	Neg	Ne
2	Faint 1+ C	1+ C	1+ C	NA	NA	NA	NA	Neg	Slide not available							
3	Faint 1+ C	1+ C granular	NA	NA	NA	NA	NA	NA	Neg	Faint 1+ C (<1	NA	NA	NA	NA	NA	NA
4	Slide not available								Slide not available							
5	Neg	1+ C granular	NA	NA	NA	NA	NA	NA	Neg	Neg	NA	NA	NA	NA	NA	NA
6	Slide not available								Slide not available							
7	Neg	1+ C granular	NA	Neg	NA	1+/2+ C	NA	NA	Neg	Neg	NA	Neg	NA	1+ C	NA	NA
8	Faint 1+ C	1+ C granular	NA	NA	1+/2+ C	NA	NA	NA	Neg	1+ C (50% cell	NA	NA	Neg	NA	NA	NA
9	Faint 1+ C	1+ C granular	NA	NA	NA	NA	Faint 1+ C	NA	NA	Neg	NA	NA	NA	NA	Neg	NA
10	Neg	1+/2+ C	NA	NA	NA	NA	NA	NA	Neg	Neg	NA	NA	NA	NA	NA	NA
11	Neg	1+ C	NA	Neg	NA	NA	NA	NA	Neg	Neg	NA	Neg	NA	NA	NA	NA
12	Slide not available								Slide not available							
13	NA	1+/2+ C gran	NA	NA	NA	NA	NA	NA	NA	Neg	NA	NA	NA	NA	NA	NA
14	NA	1+/2+ C	NA	NA	NA	NA	NA	Ost 1+ C	NA	Neg	NA	NA	NA	NA	NA	Ost neg
15	2+ C	2+ C	NA	NA	NA	NA	NA	Ost 2+ C	Neg	Neg	NA	NA	NA	NA	NA	Ost neg
16	NA	Neg/1+/2+ C	NA	NA	NA	NA	NA	NA	NA	Neg	NA	NA	NA	NA	NA	NA
17	NA	1+/2+ C gran	NA	NA	NA	NA	NA	Ost 1+ C	NA	Neg	NA	NA	NA	NA	NA	Ost neg
18	Slide not available								Slide not available							
19	NA	1+/2+ C gran	NA	NA	NA	NA	NA	NA	NA	Neg	NA	NA	NA	NA	NA	NA
20	1+ C	1+/2+ C gran	NA	NA	NA	2+ C	1+ C	NA	Neg	Neg	NA	NA	NA	2+ C	Neg	NA

Table 1: Table recapitulating HK isoforms staining in HB samples.

Referee #2 (Comments on Novelty/Model System):

Lack of in vivo animal cancer models

We thank Reviewer #2 for the useful comments.

We agree with the reviewer that an *in vivo* model would be very relevant to study metabolic aspects of HB. Currently, there are only a limited number of genetically-engineered mouse models that exist, usually displaying a mix of HCCs and HBs. For instance, an elegant model described by Mokkapati *et al.* in 2014 (Ref. 1 added at the end of this Response Letter) consisted of mice carrying a *Cited1-CreERTM-GFP* transgene coupled to *Ctnnb1ex3(Flox)* enabling tamoxifen-dependent *Ctnnb1* exon-3 deletion in hepatic stem/progenitor cells in fetal liver. At the age of 6 months, more than 90% of *Cited1-CreERTM-GFP; Ctnnb1ex3(Flox)* mice with Wnt pathway activation developed HCC and, only in a few cases, HBs. We consider that such a model or even a more specific HB model would be a great advantage for HB research; the development of such models is currently an objective of our laboratory in order to explore the oncogenic *Ctnnb1*-dependent metabolic reprogramming of HB, and the impact, in these models, of GLUT3 or LDHB blockade or deletion. However, these are long-term investigations that will require several years of development. A second option would be to inject subcutaneously immunodeficient mice with stable cell lines that are deficient for GLUT3 using shRNA or CRISPR/Cas9. Although such models would be much faster than autochthonous tumor models, they suffer from the fact that they develop from already fully transformed cell lines, injected in mice that have no immune system therefore these systems do not recapitulate the natural history of HB development. This is why we consider the elaboration of novel autochthonous models of HB for future investigations.

Referee #2 (Remarks):

The manuscript by the Meylan group compares the gene expression of different hepatoblastoma (HB) and hepatocellular carcinoma (HCC) cell lines. They validate some of the differentially expressed gene products in human biopsies: Glut3, G6PC, LDHB. Finally, they give some hints on the transcriptional regulation of Glut3 by beta catenin. Although some of the data are interesting and convincing, the manuscript lacks focus and in depth demonstration.

- 1) The relevance of Glut3 expression requires functional in vitro and in vivo studies using loss- and gain-of-function approaches*

We acknowledge this weakness in the previous version of our study. In order to give more information on the impact of GLUT3 in the different cell lines, we used siRNA, which strongly decreased its expression (~80% decrease) (Figure 7A below). Using this approach, we first found that GLUT3 expression was essential for glucose uptake (Figure 7B below) in Huh-6 but not in HepG2. Second, we performed a glycolysis assay using seahorse (Figure 7C below). Huh-6 glycolytic capacity was altered by the reduction of GLUT3 while HepG2 was insensitive to GLUT3 decrease. Finally, Hep293TT exhibited an intermediate response in absence of GLUT3 correlating

well with their partial glycolytic phenotype observed with MitoTracker and seahorse experiments. These results are now shown in the new version of the manuscript (new Figure 3D, E, F).

Figure 7: A) rt-qPCR on GLUT3 to assess the efficiency of GLUT3 silencing in the 3 hepatoblastoma cell lines. **B)** Glucose uptake assay in HepG2 and Huh-6 after transfection with siGLUT3 or siControl. **C)** Seahorse glycolytic assay on the three hepatoblastoma cell lines after transfection with siGLUT3 or siControl

The myc siRNA approach does not seem effective, as substantial residual myc expression can be observed (fig. 5A)

We agree with the reviewer that our knockdown was not effective enough. To circumvent this problem, we used a pool of 4 siRNAs, and we obtained a reduction of 80-90% of c-myc expression in HepG2. Despite strong knockdown, there was still no impact on GLUT3 expression as shown in Figure 8 below. This figure, including PFKF and HK1 gene expression, is now present in the revised manuscript as Appendix Fig S1C.

Figure 8: Real-time PCR on c-myc and GLUT3 after transfection with siMYC or siControl in HepG2 cells.

3) *Glut3*, *G6PC* and *LDHB* do not appear to reliably discriminate between tumor types as 30-50% of samples are not consistent with the rest (fig. 4C, 7I, 7J).

We agree with the reviewer that our different markers have to be considered as a complementary approach to regular H&E diagnostic but are probably not sufficient alone to discriminate between embryonal and fetal lesions. Up to now, we have not found an unambiguous marker to differentiate embryonal and fetal lesions (even if GLUT3 staining was never found in fetal tumors). The absence of a single reliable biomarker is reinforcing the concept that a combination of biomarkers will be needed for a better diagnostic. We think that our characterization of fetal and embryonal cells may pave the way to the discovery of such markers. This is now better explained in the discussion, pages 17 and 18.

Referee #3 (Remarks):

In the present study Crippa and colleagues provide an extension of the definition of hepatoblastoma (HB) subtypes, representing the most common liver cancer in children. Mainly based comparisons between cell lines representing previously established embryonal and fetal HB subtypes, as well as different hepatocellular carcinoma cell lines, representing the most common form of liver cancer in adults, the authors describe molecular and metabolic differences contributing to subtype definition. Specifically, the authors provided data suggesting a more glycolytic metabolic phenotype characteristic for the embryonal HB subtype, which was mainly attributed to TCF4/beta-Catenin-dependent induction of glucose transporter (GLUT) 3 expression. In addition, GLUT3 and other metabolic proteins were proposed as biomarkers for diagnosis of subtypes, also potentially offering novel treatment opportunities based on metabolic vulnerabilities.

The presented insights into mechanisms defining subtypes based on oncogene-dependent metabolic reprogramming are interesting and of potential translational value. Although most of the provided data are derived from in vitro experiments, some of the central findings concerning the metabolic markers have been confirmed in human HB samples. However, some of the data presented fall short of the characterization of the metabolic phenotype. Also, some of the experimental results were confusing and some of the figures did not have clear labeling/figure legends, making it difficult to interpret the data. Therefore, the study at this stage requires revision addressing the following points:

Major Points:

Fig. 2E: It is surprising that while that the authors provided Seahorse data, they seemingly did not exploited the full potential of the method concerning the metabolic phenotyping of the cells. The authors should measure extracellular acidification in the course of the so-called "glycolysis stress test", which includes sequential addition of glucose, oligomycin (assessing maximum glycolytic capacity) and 2-DG (inhibition of glycolysis), drawing a much clearer picture of the glycolytic phenotype in comparison to the cell line comparisons under basal conditions only.

First, we would like to thank Reviewer #3 for the very useful comments and suggestions.

We agree with the reviewer that we mainly studied the glycolytic phenotype at the basal level. We, therefore, decided to follow the reviewer's recommendation in order to improve our understanding of the metabolic capacity of the different cell lines. Thus, we performed a seahorse glycolytic stress assay after 24 hours of glucose starvation. Glucose was then added during the seahorse experiment followed by the addition of 2-DG. Unfortunately, in our hands, cells did not respond to oligomycin at the different concentrations we tried so we did not add it here. We found that Huh-6 exhibited the highest glucose response in comparison to all other cell lines (Figure 1bis below). Interestingly we also observed a response to glucose in Hep293TT confirming the glycolytic potential that we found with MitoTracker. We did not observe a clear response to glucose in the other tested cell lines. We added this result as new Figure 2F in the revised version of the manuscript.

Figure 1bis: After 24 hours starvation, 10 mM glucose were added to the well followed by 2-DG to block glycolysis at a concentration of 50mM. Sequential measurements of ECAR after the different injections were realized.

For selected cell lines (e.g. Huh-6 vs Hep293TT) Seahorse glycolysis assays (as described above) or alternatively glucose uptake assays should be performed in the presence and absence of *Glut3* and/or *beta-catenin* knockdown, thereby providing more robust evidence for the central role of *beta-catenin* and/or *Glut3* for the "sugar metabolic phenotype" of the embryonal-like subtype of HB.

We acknowledge this weakness in the previous version of our study. In order to give more information on the impact of GLUT3 in the different cell lines, we used siRNA on GLUT3 (Figure 7A bis below). As a result, we found that GLUT3 expression was essential for glucose uptake (Figure 7B bis below) in Huh-6 but not in HepG2. Secondly, we performed a glycolysis assay using seahorse under siGLUT3 context (Figure 7C bis below). Huh-6 glycolytic capacity was altered by the decreased expression of GLUT3 while HepG2 were insensitive to its loss. Finally, Hep293TT exhibited an intermediate response in absence of GLUT3 correlating well with their partial glycolytic phenotype observed by mitotracker and seahorse experiments. These data are now present in the new version of the manuscript (Figure 3 D, E, F).

Figure 7bis: A) rt-qPCR on GLUT3 to assess the efficiency of GLUT3 silencing in the 3 hepatoblastoma cell lines. B) Glucose uptake assay in HepG2 and Huh-6 after transfection with siGLUT3 or siControl. C) Seahorse glycolytic assay on the three hepatoblastoma cell lines after transfection with siGLUT3 or siControl

Minor Points:

Fig. 2A: It is unclear from which material the presented RNAseq data were derived from. Does HB or HCC refer to a specific cell line of the respective type or do the gene expression levels represent mean values of several cell lines? Please specify accordingly

We performed RNA sequencing on 4 HB cell lines and 4 HCC cell lines. In the different graphs, HB and HCC refer to the average of all HB cells compared to the average of all HCC cells. This is now better explained in the text page 27 and in the Figure legend.

While the Supplemental tables as excel sheets are informative, the provided PDF formats are not useful and should be deleted.

During the online submission process, PDF files were automatically generated from our Excel files, which we regret but cannot control. We take this opportunity to mention that we add another Supplementary Table on the differentially expressed genes between fetal-like cells and embryonal-like cells called Table EV2 in the revised version of the manuscript.

Fig5A; page9 line 15-17: Myc expression is more strongly reduced upon knockdown of beta-catenin than in the samples with direct Myc KD, which was obviously not very efficient. This is questioning whether these data really allow the conclusion that Myc does not exert any effect on Glut3. Also, why did the authors not include the corresponding Myc knockdown samples in the corresponding analysis shown in Fig. S4A. Please comment.

We agree with the reviewer (and reviewer #2 before) that our c-myc knock down was not satisfactory. We have now used a pool of 4 siRNAs allowing a reduction of 80-90% of c-myc expression in HepG2, which still did not affect GLUT3 or PFKP expression as shown in Figure 9 below. In HepG2, HK1 and PFKP are lowly expressed but we still detected less HK1 upon c-myc silencing. The low level of HK1 expression in HepG2 is questioning about the relevance of this decrease. Finally, LDHB was not even detectable in this experiment. This figure is present in the revised version of the manuscript as Appendix Figure S1C.

Figure 9: Real-time PCR on c-myc, GLUT3, PFKP, HK1 after transfection with siMYC or siControl in HepG2 cells.

Fig.6, page 12; discussion page 15: How do elevated levels of HK2 (also reflected at the level of higher sensitivity to 3BP) relate to reduced glycolytic activity in HB-fetal lines vs elevated glycolysis

in embryonal-like HK1-induced cells? Please add a comment to the functional consequences of different HK isoforms in the different subtypes with respect to the glycolytic phenotype.

There exists four hexokinase isoforms that catalyze the first step of glucose metabolism, which is the ATP-dependent phosphorylation of glucose to G6P. It was described that the level of G6P can regulate this process by inhibiting HKs as negative feedback, though inorganic phosphate (P_i) and then block metabolism (Ref. 2 at this end of this Response Letter). However, unlike HK2 and HK3, HK1 itself is not directly regulated by P_i , which better suits its ubiquitous catabolic role and might allow a higher glycolytic activity because of the absence of negative feedback.

Information related to HB surgical specimens is missing in the Material and Methods section.

The HB specimens were provided by two hospitals. 13 cases from the Division of Clinical Pathology, Geneva University Hospitals, Switzerland and 20 from the Department of Pathology, Hôpital Bicêtre, HUPS, Assistance Publique-Hôpitaux de Paris, INSERM U1193, Faculté de Médecine Université Paris Sud, France. As those archived samples were used only for IHC we did not need any ethical authorization. This information has now been added in the Appendix Supplementary text , page 4.

References:

1: Mokkaapati S, Niopek K, Huang L, Cunniff KJ, Ruteshouser EC, deCaestecker M, Finegold MJ, Huff V. β -catenin activation in a novel liver progenitor cell type is sufficient to cause hepatocellular carcinoma and hepatoblastoma. *Cancer Res.* 2014 Aug 15;74(16):4515-25. doi: 10.1158/0008-5472.CAN-13-3275. Epub 2014 May 21. PubMed PMID: 24848510; PubMed Central PMCID: PMC4134699.

2: Aleshin AE, Zeng C, Bourenkov GP, Bartunik HD, Fromm HJ, Honzatko RB. The mechanism of regulation of hexokinase: new insights from the crystal structure of recombinant human brain hexokinase complexed with glucose and glucose-6-phosphate. *Structure.* 1998 Jan 15;6(1):39-50. PubMed PMID: 9493266.

2nd Editorial Decision

14 August 2017

Thank you for the submission of your revised manuscript to EMBO Molecular Medicine. We have now received the enclosed reports from the reviewers that were asked to re-assess it. As you will see, while reviewers 1 and 3 are now supportive, #2 is still critical based on the lack of in vivo experimentation and because in his/her opinion, the identified markers cannot be used in the clinic. As for the first point, reviewers 1 and 3 are less critical (although #3 is clearly not enthusiastic about this aspect).

I am prepared to accept your manuscript for publication pending a rebuttal on both points raised by reviewer 2, Based on your reply and the addition of appropriate comments in the manuscript on the limitations of the study in these respects, I am prepared to make an editorial decision on your next, final version of the manuscript.

Please also comply with the following editorial requirements:

- 1) Please change the reference list formatting from 10 authors et al. to 20 authors et al.
- 2) Please remove all appendix figure legends from the Appendix and include them as individual tabs in the excel files
- 3) Please name the Appendix file "Appendix" and note that you will need to name and refer to Appendix items in the main text and the Appendix (including the TOC page) as follows: Appendix Figure S1, Appendix Table S1, Appendix Supplementary Methods, etc. The Appendix will not be edited post-acceptance and will thus be published as is.

4) Please consider changing the EV tables into EV datasets and update callouts in the manuscript accordingly.

5) I note that there are a few issues with some figures. Firstly, the quality of some images is not ideal, e.g. Fig. 2B, Fig. EV1C. Also, many figures would benefit from increased text and graph size (Fig. 1C, Appendix fig S4 B & D) and more visible scale bars (Fig 4 A & B, Fig 7A-I, Fig. EV2A-F). Fig. 4B would perhaps benefit from naming the panels somehow, including in the legend. Finally, please provide a scale bar for Fig. EV1C

6) We encourage the publication of source data, with the aim of making primary data more accessible and transparent to the reader. Would you be willing to provide a PDF file per figure that contains the original, uncropped and unprocessed scans of all or at least the key gels used in the manuscript and/or source data sets for relevant graphs? The files should be labeled with the appropriate figure/panel number, and in the case of gels, should have molecular weight markers; further annotation may be useful but is not essential. The files will be published online with the article as supplementary "Source Data" files. If you have any questions regarding this just contact me.

7) Every published paper includes a 'Synopsis' to further enhance discoverability. Synopses are displayed on the journal webpage and are freely accessible to all readers. They include a short description as well as 2-5 one-sentence bullet points that summarise the key NEW findings of the paper. The bullet points should be designed to be complementary to the abstract - i.e. not repeat the same text. We encourage inclusion of key acronyms and quantitative information. Please use the passive voice. Please attach this information in a separate file or send them by email, we will incorporate it accordingly. We also encourage the provision of striking image or visual abstract to illustrate your article. If you do, please provide a jpeg file 550 px-wide x 400-px high.

For all the above, please refer to our author guidelines (<http://embomolmed.embopress.org/authorguide>).

Please submit your revised manuscript within two weeks. I look forward to seeing a revised form of your manuscript as soon as possible.

***** Reviewer's comments *****

Referee #1 (Comments on Novelty/Model System):

I went through the new version of the manuscript, as well as the rebuttal to my previous critiques and suggestions. Overall I think that the authors have adequately addressed my concerns. I think that the manuscript merits publication as it is.

I also looked to the others reviewers comments. I think that the mouse model that they ask for is compensated by the human data that the authors show.

Referee #1 (Remarks):

The manuscript has been improved and the authors have addressed all my concerns and suggestions.

Referee #2 (Remarks):

The authors addressed some of the issues previously raised. However they did not provide in vivo evidence on animal models of cancer. In addition, the markers they identified cannot be used to unambiguously classify human tumor biopsies. In my opinion, these two issues are major flaws for the high standard of publication in EMM.

Referee #3 (Comments on Novelty/Model System):

The technical quality clearly improved upon revision. The reason for still rating it medium is mainly

due to the lack of *in vivo* approaches.

Referee #3 (Remarks):

The authors addressed all the points that I raised in my original review. They now provide a more extensive and appropriate analysis of the metabolic phenotypes using Seahorse assays, which support their main hypothesis. Also, the effects of Glut3 knockdown were in line with the notion that Glut3 expression in hepatoblastoma cell lines contributes to glycolytic phenotype.

2nd Revision - authors' response

16 August 2017

Referee #2 (Remarks):

They did not provide in vivo evidence on animal models of cancer.

We agree with the reviewer that there is a lack of animal models in this study.

Two principal *in vivo* experimental approaches exist to interrogate the biology of HB. The first one relies on human tumor cell line xenotransplantation into immunodeficient animals. Although this approach is fast, it does not recapitulate the complex and dynamic cellular communications between tumor cells and immune cells of the tumor microenvironment. In addition, these models rely on the growth of already fully transformed cell lines, failing to mirror the steps of tumor initiation. The second approach relies on the use of genetically engineered mice. In such models, tumor initiation through to tumor progression occurs in an immunocompetent host. For hepatoblastoma, a limited number of mouse models have been described: these include transgenic *LAP-tTA (Tet-OFF)*; *TRE-Myc* enabling liver specific expression of *Myc* oncogene (PMID: 15475948), *ApoE-rtTA (Tet-ON)*; *TRE-LIN28B* enabling liver specific expression of *LIN28B* oncogene (PMID: 25117712) and *Cited1-CreERTM-GFP* transgene coupled to *Ctnnb1ex3(Flox)* enabling tamoxifen-dependent *Ctnnb1* exon-3 deletion in hepatic stem/progenitor cells in fetal liver (PMID: 24848510). In the latter, mice mostly developed HCC and, only in a few cases, HBs.

In our study, we concentrated our efforts on deciphering a new metabolic classification of HB, based on human tumor cell lines and a large collection of tumor tissue specimens. For this very rare disease, because of the limited number of *in vivo* approaches available as mentioned above, we believe our work could constitute a foundation to refine mouse models of human HB, based on activation of oncogene activation coupled to deregulation of the expression of metabolic genes. We hope this will offer a new avenue to generate mouse models that develop only HB and not HCC, and more specifically – depending on perturbation in metabolic pathway – that develop distinct histological subtypes, i.e. fetal or embryonal HB. These long term projects culminating in the generation of new and more precise models of human HB will enable preclinical trials to evaluate new treatment opportunities for children suffering from this malignancy.

In the discussion section of our manuscript, two paragraphs before the end, we discuss the limitations of our study and perspectives to refine mouse models in future studies.

In addition, the markers they identified cannot be used to unambiguously classify human tumor biopsies.

We agree that our new biomarkers, GLUT3, LDHB and G6PC, used in our study to stain human HB sections, nicely illustrate distinct expression patterns between fetal and embryonal components of HB, but are currently not able alone to replace current histology for diagnosis. This is why, in the discussion, we state that they could be used as future potential complementary tools to regular histopathology. Most importantly, we think our study, which for the first time highlights two different metabolic subtypes of human HB, will incentivize new investigations aiming to discover and test additional metabolic proteins. Hopefully, these future investigations will collectively advance our capabilities to unambiguously diagnose HB.

In the discussion section of our manuscript, at the last paragraph, we discuss the limitations of our study in that aspect and how future investigations could help for clinical diagnosis.

Corresponding Author Name: Meylan
Journal Submitted to: EMBO Molecular Medicine
Manuscript Number: EMM-2017-07814